# Human Fis1 regulates mitochondrial dynamics through inhibition of the fusion machinery

Rong Yu[1], Shao-Bo Jin[2], Urban Lendahl[2], Monica Nistér[1,*,†] (ID) & Jian Zhao[1,†,**] (ID)

## Abstract

Mitochondrial dynamics is important for life. At center stage for mitochondrial dynamics, the balance between mitochondrial fission and fusion is a set of dynamin-related GTPases that drive mitochondrial fission and fusion. Fission is executed by the GTPases Drp1 and Dyn2, whereas the GTPases Mfn1, Mfn2, and OPA1 promote fusion. Recruitment of Drp1 to mitochondria is a critical step in fission. In yeast, Fis1p recruits the Drp1 homolog Dnm1p to mitochondria through Mdv1p and Caf4p, but whether human Fis1 (hFis1) promotes fission through a similar mechanism as in yeast is not established. Here, we show that hFis1-mediated mitochondrial fragmentation occurs in the absence of Drp1 and Dyn2, suggesting that they are dispensable for hFis1 function. hFis1 instead binds to Mfn1, Mfn2, and OPA1 and inhibits their GTPase activity, thus blocking the fusion machinery. Consistent with this, disruption of the fusion machinery in Drp1$^{-/-}$ cells phenocopies the fragmentation phenotype induced by hFis1 over-expression. In sum, our data suggest a novel role for hFis1 as an inhibitor of the fusion machinery, revealing an important functional evolutionary divergence between yeast and mammalian Fis1 proteins.

**Keywords** Drp1; hFis1; mitochondrial dynamics; mitofusins; OPA1
**Subject Categories** Membrane & Intracellular Transport
The EMBO Journal (2019) 38: e99748

See also: **M Liesa et al** (April 2019)

## Introduction

Mitochondria are dynamic organelles that constantly alter their shape in response to changes in cellular physiological conditions or as a result of pathophysiology. The mitochondrial network is determined through a balance between mitochondrial fission and fusion events, collectively referred to as mitochondrial dynamics, which produces fragmented or fused mitochondria, respectively and is closely associated with mitochondrial function (Chan, 2012; Schrepfer & Scorrano, 2016). Mitochondrial fragmentation can be caused by stimulation of fission activity and/or inhibition of mitochondrial fusion. Conversely, inhibition of fission and/or stimulation of fusion leads to mitochondrial elongation. An aberrant balance between fission and fusion has been observed in a number of human diseases and impacts a broad range of cellular functions (Ong & Hausenloy, 2010; Romanello et al, 2010; Yoon et al, 2011; Rehman et al, 2012; Youle & van der Bliek, 2012; Schrepfer & Scorrano, 2016; Chen & Chan, 2017; Trotta & Chipuk, 2017).

Mitochondria are evolutionarily derived from the endosymbiosis of alpha-proteobacteria, which likely was a unique event in the history of eukaryotic evolution. The mitochondrial division machinery probably arose considerably later, but there are indeed a number of highly conserved proteins regulating mitochondrial dynamics in yeast and mammals (Okamoto & Shaw, 2005; Zhao et al, 2013; Kraus & Ryan, 2017; Ramachandran, 2018), indicating that at least some of the fundamental mechanisms controlling mitochondrial dynamics have a long evolutionary history. Several of these proteins are related to dynamin, a GTPase which pinches off vesicles from the cell surface (Pagliuso et al, 2018). Thus, in mammals, mitochondrial fission is executed by the GTPases dynamin-related protein 1 (Drp1) and dynamin-2 (Dyn2) (Smirnova et al, 1998; Lee et al, 2016). Drp1 is primarily distributed in the cytosol and translocated to the mitochondrial surface during mitochondrial division, where it assembles into higher-order complexes at endoplasmic reticulum (ER)–mitochondrial contact sites to wrap around the mitochondria inducing mitochondrial fission via its GTPase activity. The final abscission step is carried out by Dyn2 (Otera et al, 2013; Friedman & Nunnari, 2014; Lee et al, 2016; Mishra & Chan, 2016; Prudent & McBride, 2016; Kraus & Ryan, 2017). Mitochondrial fusion is regulated by three other dynamin-related GTPases: Mitofusins (Mfn1 and Mfn2), which are located in the mitochondrial outer membrane (MOM) and optic atrophy 1 (OPA1), located in the mitochondrial inner membrane (MIM) (Willems et al, 2015; Schrepfer & Scorrano, 2016). Mfn1 and Mfn2 are essential for fusion of mitochondrial outer membranes, whereas the mitochondrial inner membrane-associated OPA1 is required for fusion of the inner membranes (Willems et al, 2015; Schrepfer & Scorrano, 2016; Pagliuso et al, 2018).

1 Department of Oncology-Pathology, Karolinska Institutet, Karolinska University Hospital Solna, Stockholm, Sweden
2 Department of Cell and Molecular Biology, Karolinska Institutet, Stockholm, Sweden
   *Corresponding author. Tel: +46 8 51770309; E-mail: monica.nister@ki.se
   **Corresponding author. Tel: +46 8 51770585; E-mail: jian.zhao@ki.se
   †These authors contributed equally to this work as senior authors

The pro-fission GTPase Drp1 does not contain a membrane-localizing pleckstrin homology (PH) domain, transmembrane (TM), or any other membrane-anchoring domain and therefore needs to be actively recruited to the mitochondrial surface by MOM-anchored receptors in order to execute its function. In yeast, recruitment of Dnm1p (the yeast Drp1 homolog) to mitochondria is carried out by the evolutionarily conserved membrane-anchored protein Fis1p (the yeast Fis1 homolog) through interaction with Mdv1p and Caf4p, promoting mitochondrial fission (Hoppins *et al*, 2007). As Fis1 is evolutionarily conserved from yeast to humans, mammalian Fis1 is believed to have a similar role to its homolog Fis1p in yeast, i.e., to recruit Drp1 to mitochondria and promote mitochondrial fission. In keeping with a pro-fission function, overexpression of Fis1 causes extensive mitochondrial fragmentation and depletion of Fis1 causes mitochondrial elongation (James *et al*, 2003; Yoon *et al*, 2003; Stojanovski *et al*, 2004; Jofuku *et al*, 2005; Yu *et al*, 2005). Interestingly, while Mitofusins, OPA1, Drp1, Dyn2, and Fis1 are highly conserved between yeast and mammals, Mdv1p and Caf4p lack obvious mammalian homologs (Okamoto & Shaw, 2005; Hoppins *et al*, 2007; Bui & Shaw, 2013; Zhao *et al*, 2013). Instead, mammalian cells contain a set of proteins, Mff and MIEF1/2 (MiD51/49), which are anchored in the mitochondrial outer membrane and serve to mediate recruitment of Drp1 to the mitochondrial surface (Gandre-Babbe & van der Bliek, 2008; Otera *et al*, 2010; Palmer *et al*, 2011; Zhao *et al*, 2011; Koirala *et al*, 2013; Liu *et al*, 2013; Loson *et al*, 2013; Yu *et al*, 2017). Furthermore, increased or decreased levels of Fis1 do not seem to regulate the amount of Drp1 at mitochondria in mammalian cells (Suzuki *et al*, 2003; Lee *et al*, 2004), and according to several reports, Fis1 is largely dispensable for Drp1 recruitment to mitochondria and Drp1-mediated fission (Otera *et al*, 2010, 2013; Bui & Shaw, 2013; Koirala *et al*, 2013; Loson *et al*, 2013; Osellame *et al*, 2016).

Given the absence of Mdv1p and Caf4p in mammals, the actual role of hFis1 in the mitochondrial fission process remains enigmatic (Okamoto & Shaw, 2005; Hoppins *et al*, 2007; Bui & Shaw, 2013; Otera *et al*, 2013; Zhao *et al*, 2013). Some reports argue that mammalian Fis1 interacts with Drp1 and can serve as a receptor for Drp1 (James *et al*, 2003; Yoon *et al*, 2003), whereas other studies reveal that mammalian Fis1 is largely dispensable for Drp1 recruitment (Suzuki *et al*, 2003; Lee *et al*, 2004). In this study, we address the functional role of human Fis1 (hFis1). We show that both Drp1 and Dyn2 are dispensable for hFis1-mediated mitochondrial fragmentation. hFis1 instead interacts robustly with Mfn1, Mfn2, and OPA1, and overexpression of hFis1 blocks their GTPase activity, leading to reduced mitochondrial fusion and shifting the balance of mitochondrial dynamics toward fission. In conclusion, our data suggest a novel function for hFis1 in regulating mitochondrial dynamics in mammals and reveal an evolutionary divergence of Fis1 function between yeast and mammals.

# Results

### Drp1 and Dyn2 are largely dispensable for hFis1-mediated mitochondrial fragmentation

We first assessed whether Fis1-induced fragmentation was dependent on the presence of Drp1. Overexpression of human Fis1 (hFis1) triggered extensive mitochondrial fragmentation in wild-type (WT) 293T cells (i.e., in the presence of endogenous Drp1), resulting in small and punctuate mitochondria in most ($92 \pm 0.4\%$) cells compared to empty vector control ($1.5 \pm 2.3\%$) (Fig 1A, upper panel; summarized in Fig 1E), in line with several previous reports (James *et al*, 2003; Yoon *et al*, 2003; Stojanovski *et al*, 2004; Yu *et al*, 2005; Alirol *et al*, 2006). To explore the role of Drp1 in hFis1-mediated fragmentation, we next generated a DRP1-deficient 293T cell line (Drp1$^{-/-}$) using CRISPR/Cas9-mediated gene editing (Appendix Fig S1), and ablation of Drp1 as expected halted mitochondrial fission, resulting in a super-fused tubular mitochondrial network (Fig 1A, lower left panel). Overexpression of hFis1, however, still efficiently induced mitochondrial fragmentation in the Drp1$^{-/-}$ 293T cells ($85.9 \pm 1.2\%$), although there was a slight decrease in the number of cells with fragmented mitochondria in comparison with WT 293T cells (Fig 1A, lower right panel; summarized in Fig 1E). These results indicate that Drp1 is largely dispensable for mitochondrial fragmentation induced by hFis1.

To further investigate whether hFis1-induced fragmentation could also occur in other types of human cells in the absence of endogenous Drp1, we generated a DRP1-deficient (Drp1$^{-/-}$) HeLa cell line using CRISPR/Cas9-mediated gene editing (Appendix Fig S2). Similarly, this led to a super-fused tubular mitochondrial network (Appendix Fig S2D), and hFis1 overexpression still triggered mitochondrial fragmentation in Drp1$^{-/-}$ HeLa cells ($38.8 \pm 2.3\%$) (Fig EV1). Overall, this confirms that hFis1 can promote mitochondrial fragmentation in the absence of Drp1, but loss of Drp1 partially reduces hFis1-induced fragmentation.

While hFis1-induced fragmentation occurred also in the absence of Drp1, there were some noticeable differences between overexpression of hFis1 in wild-type (control) and Drp1$^{-/-}$ (deficient) cells: The size of fragmented (punctate) mitochondria was larger with an average size $\sim 0.48 \pm 0.01 \, \mu m^2$ in Drp1$^{-/-}$ cells compared to an average size of $\sim 0.28 \pm 0.01 \, \mu m^2$ in WT 293T cells. At the same time, the number of mitochondria was lower in Drp1-deficient cells (Fig 1B and C), i.e., mitochondria were more fragmented in WT cells, whereas most mitochondria in Drp1$^{-/-}$ cells appeared as larger spheres. A similar phenotype was also observed in Drp1$^{-/-}$ HeLa cells expressing Myc-hFis1 (Fig EV1). These subtle differences in mitochondrial phenotype may be attributed to the continuously ongoing Drp1-mediated fission occurring in WT but being blocked in Drp1$^{-/-}$ cells.

To further elaborate on the role of hFis1 in mitochondrial dynamics, we generated several hFis1 mutants (Fig 1D) and tested their effects on mitochondrial morphology in WT and Drp1$^{-/-}$ 293T cells. As previously reported (Yoon *et al*, 2003; Stojanovski *et al*, 2004), the deletion mutant Myc-hFis1$^{\Delta TM/C}$, lacking the TM domain and the C-terminal tail (from residues 123–152), was diffusely distributed in the cytosol and had no effect on mitochondrial morphology in WT and Drp1$^{-/-}$ 293T cells (Figs 1E and EV2A). In contrast, GFP-hFis1$^{\Delta 1-121}$ (lacking the cytosolic domain of hFis1, i.e., consisting of GFP fused to the hFis1 TM domain and C-terminal tail including residues 122–152) was still localized to the mitochondrial surface (Stojanovski *et al*, 2004) and triggered mitochondrial aggregation in both WT ($47 \pm 3.1\%$) and Drp1$^{-/-}$ 293T cells ($51.1 \pm 4.1\%$) (Figs 1E and EV2A). These data indicate that both the cytosolic domain and the C-terminal region including the TM domain and C-terminal tail are required for hFis1-induced

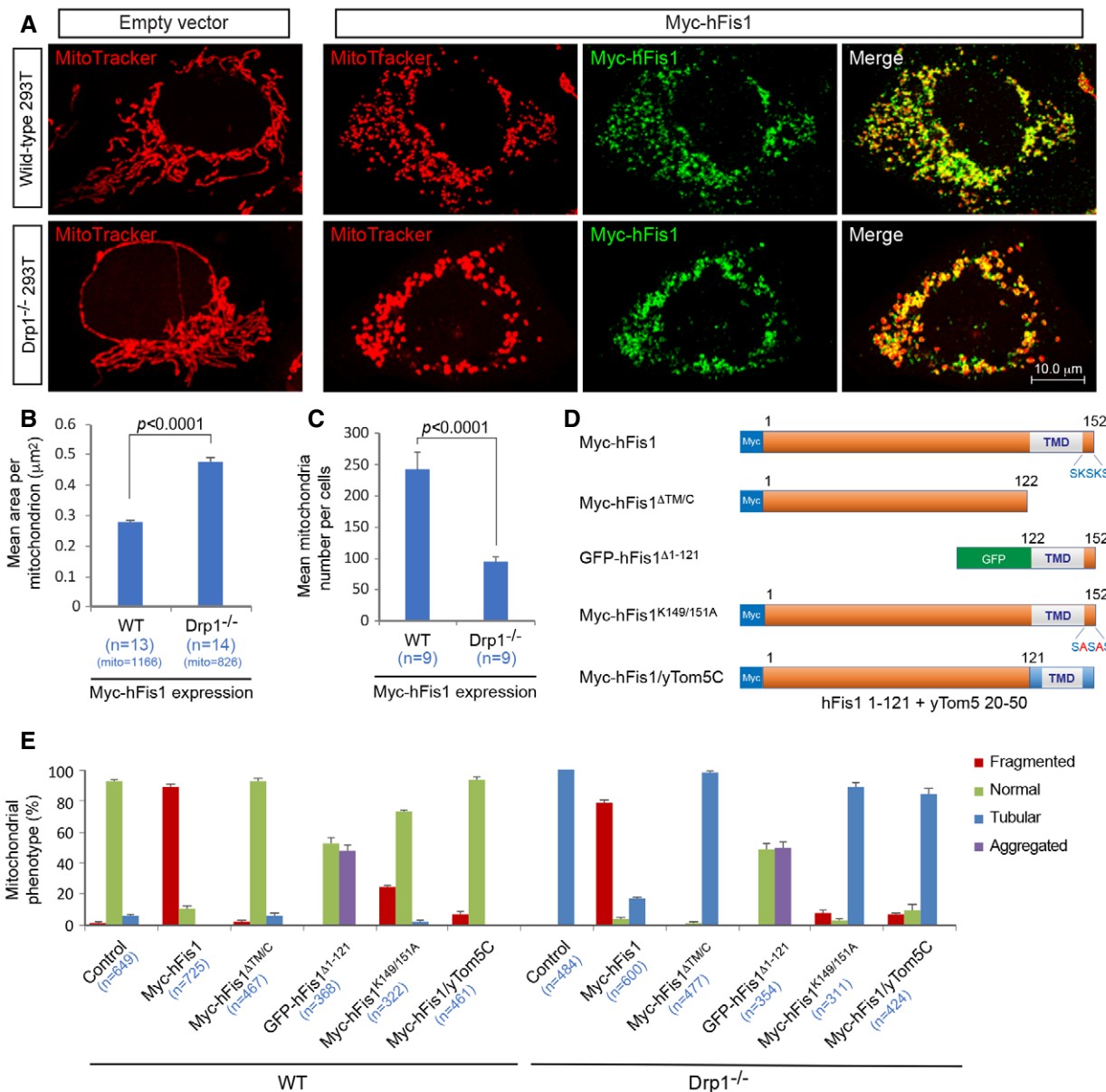

**Figure 1. Drp1 is not required for hFis1-induced mitochondrial fragmentation.**

A   Confocal images of mitochondrial morphology in wild-type (WT) and Drp1$^{-/-}$ 293T cells transfected with empty vector (left panel) and Myc-hFis1 (right panel), stained with MitoTracker (red) followed by immunostaining with anti-Myc antibody (green).
B   Quantitative analyses of fragmented mitochondria size (mean area ($\mu$m$^2$) per mitochondrion) after Myc-hFis1 overexpression in WT and Drp1$^{-/-}$ 293T cells using ImageJ software (Particle analysis) in three independent experiments. In each cell, only dispersed individual mitochondria were analyzed. The total number of mitochondria (mito) analyzed is indicated for each condition.
C   Quantitative analysis of mean mitochondria number per cell in WT and Drp1$^{-/-}$ 293T cells overexpressing Myc-hFis1 using Image J software (Particle analysis) in three independent experiments.
D   Schematic representation of hFis1 mutants used in this experiment.
E   Percentages of cells with indicated mitochondrial morphologies in WT and Drp1$^{-/-}$ 293T cells transfected with empty vector (control), Myc-hFis1 and either Myc- or GFP-tagged mutants as indicated in three independent experiments.

Data information: Data are expressed as means $\pm$ SEM and were statistically analyzed by Student's *t*-test. *n* represents the number of cells analyzed (B, C, and E).

---

mitochondrial fragmentation regardless of whether Drp1 is present or not in cells. We further evaluated the effects of the ER (endoplasmic reticulum) mistargeting mutant Myc-hFis1$^{K149/151A}$ (i.e., full-length hFis1 carrying lysine to alanine mutations in the residues 149

and 151) (Stojanovski *et al*, 2004; Delille & Schrader, 2008) on mitochondrial morphology. When expressed in WT and Drp1$^{-/-}$ 293T cells, Myc-hFis1$^{K149/151A}$ significantly reduced the number of cells with fragmented mitochondria to 24.2 $\pm$ 0.8% in WT cells and to

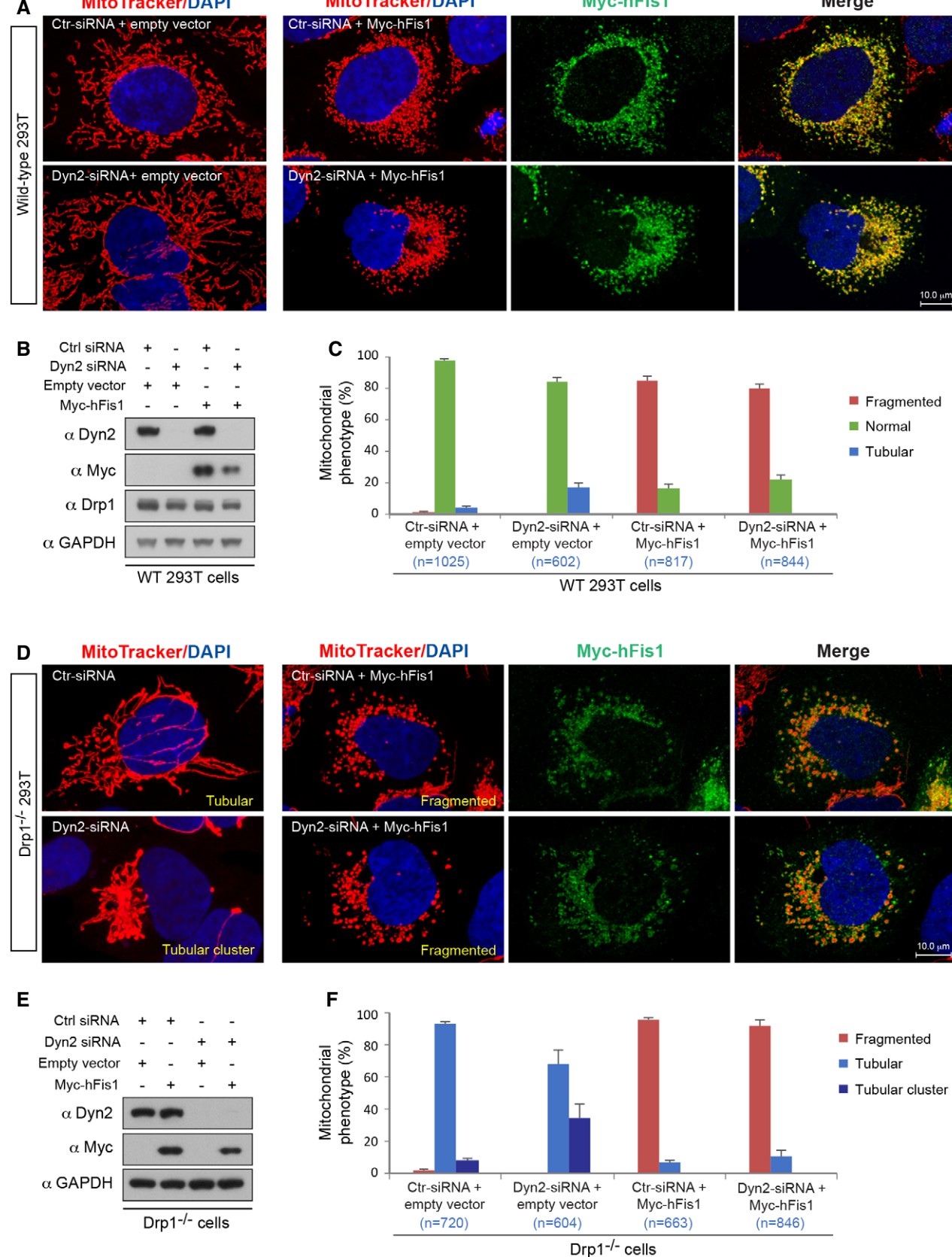

**Figure 2.**

◀

**Figure 2. Dyn2 is not required for hFis1-induced mitochondrial fragmentation.**

A   Confocal images of mitochondrial morphology in WT 293T cells treated with scrambled siRNA (control) and Dyn2 siRNA as indicated, followed by transfection with empty vector (left panel) and Myc-hFis1 (right three panels) as indicated. Cells were stained with MitoTracker (red) followed by immunostaining with anti-Myc antibody (green).

B   Western blot of Dyn2, Myc-hFis1, Drp1, and GAPDH in WT 293T cells collected from the experiments in (A).

C   Percentages of cells with indicated mitochondrial morphologies in WT 293T cells treated with scrambled siRNA and Dyn2 siRNA as indicated, followed by transfection with empty vector or Myc-hFis1 in three independent experiments.

D   Confocal images of mitochondrial morphology in Drp1$^{-/-}$ 293T cells treated with scrambled siRNA and Dyn2 siRNA as indicated, followed by transfection with empty vector (left panel) and Myc-hFis1 (right three panels). Cells were stained with MitoTracker (red) followed by immunostaining with anti-Myc antibody (green). Representative examples of tubular, fragmented, and tubular cluster phenotypes are indicated.

E   Western blot of Dyn2, Myc-hFis1, and GAPDH in Drp1$^{-/-}$ 293T cells collected from the experiments in (D).

F   Percentages of cells with indicated mitochondrial morphologies in Drp1$^{-/-}$ 293T cells treated with scrambled siRNA or Dyn2 siRNA, followed by transfection with empty vector or Myc-hFis1 as indicated in three independent experiments.

Data information: Data are expressed as means ± SEM and were statistically analyzed by Student's *t*-test. *n* represents the number of cells analyzed (C and F). Source data are available online for this figure.

7.9 ± 1.6% in Drp1$^{-/-}$ cells (Figs 1E and EV2A). To further explore the function of the TM domain and C-terminal tail of hFis1, we generated a hFis1/yTom5C mutant, in which the TM and C-terminal tail domains of hFis1 were replaced with the corresponding regions of the mitochondrial C-tail-anchored yeast protein Tom5, as previously reported (Jofuku *et al*, 2005). Expression of this construct largely abolished hFis1-induced mitochondrial fragmentation in both WT and Drp1$^{-/-}$ 293T cells (Figs 1E and EV2A), implying that hFis1 fission function was lost in spite of the fact that the mutant mainly localized on mitochondria (Fig EV2A). In summary, these data establish that intact N-terminal cytosolic as well as C-terminal TM and tail domains are all required for hFis1-mediated mitochondrial fragmentation to occur both in the presence and in the absence of Drp1.

To extend the observations on the mitochondrial phenotype, we assessed the consequences of hFis1 overexpression on apoptosis and autophagy in WT 293T and Drp1$^{-/-}$ 293T cells. As shown in Fig EV2B and C, overexpression of WT hFis1 and mutants did not increase the amounts of cleaved PARP or LC3B-II, confirming that the observed hFis1-induced mitochondrial fragmentation is not an unspecific effect resulting from apoptosis or autophagy that is potentially associated with excessive expression of mitochondrial proteins.

Dynamin-2 (Dyn2) has recently been reported to regulate the final abscission step of mitochondrial fission after Drp1 recruitment and polymerization (Lee *et al*, 2016). We therefore tested whether Dyn2 is required for hFis1-mediated mitochondrial fragmentation. In WT 293T cells, depletion of Dyn2 by small interfering RNA (siRNA) resulted in a moderately elongated mitochondrial network in 16.5 ± 2.7% of cells compared to control (3.4 ± 0.8%, *P* = 0.0006) (Fig 2A–C), in line with previous observations (Lee *et al*, 2016). Overexpression of hFis1 in Dyn2-deficient cells still caused extensive mitochondrial fragmentation in 78.9 ± 2.8% of cells, comparable to the effect of hFis1 overexpression in WT 293T cells (84.1 ± 3.0%) (Fig 2A–C), indicating that Dyn2 is also dispensable for hFis1-mediated fragmentation. Moreover, depletion of Dyn2 by siRNA in Drp1$^{-/-}$ 293T cells did not prevent mitochondrial fragmentation induced by hFis1 overexpression in most of cells (90.6 ± 3.5%) compared to controls (94.7 ± 1.7%) (Fig 2D–F), indicating that simultaneous depletion of Drp1 and Dyn2 does not prevent hFis1-induced fragmentation either. Taken together, these data show that hFis1 can induce mitochondrial fragmentation under Drp1/Dyn2-deficient conditions.

**Mfn1, Mfn2, and OPA1 are major hFis1-binding proteins**

The data described above indicate that mitochondrial fission can occur in a Drp1- and Dyn2-dispensable manner and that hFis1 plays a role as an alternative fission mediator. To assess the role of hFis1 further, we examined whether hFis1 instead influenced the mitochondrial pro-fusion machinery. Depletion of hFis1 by siRNA in Drp1$^{-/-}$ 293T cells altered mitochondrial morphology, resulting in a more super-fused tubular clustering phenotype of mitochondria localized asymmetrically to one side of the nucleus (Fig 3A). This phenotype was quite reminiscent of the phenotype seen in Drp1$^{-/-}$ cells overexpressing either of the pro-fusion GTPases Mfn1, Mfn2, or OPA1 (Fig 3B and C).

These results may imply a functional link between hFis1 and the pro-fusion machinery. We first addressed this by examining the interaction of hFis1 with the pro-fission and pro-fusion GTPases. A weak interaction between endogenous hFis1 and Drp1 was observed following *in vivo* chemical crosslinking (Hajek *et al*, 2007; Zhao *et al*, 2011) before co-immunoprecipitation (co-IP) (Fig 3D), in agreement with previous studies (Yoon *et al*, 2003; Yu *et al*, 2005). No interactions were detected between hFis1 and Dyn2, Miro1 (a MOM-anchored protein) (Fransson *et al*, 2003) or MTGM (a MIM-anchored protein, also known as Romo1) (Chung *et al*, 2006; Zhao *et al*, 2009), irrespective of whether Drp1 was present or not (Fig 3D). In contrast, hFis1 robustly interacted with Mfn1, Mfn2, and OPA1 at endogenous levels under conditions of chemical crosslinking (Fig 3D). To further assess whether the interaction of hFis1 with Mfn1, Mfn2, and OPA1 was transient or stable, we also performed co-IP in the absence of chemical crosslinking. hFis1 still efficiently bound to Mfn1, Mfn2, and OPA1 at endogenous levels to form stable protein complexes, whereas no interactions were detected between hFis1 and either Drp1, Dyn2, Miro1, or MTGM (Fig 3E). A similar interaction pattern at endogenous levels was obtained by co-IP in HeLa cells without chemical crosslinking (Fig 3F). Together, these data indicate that hFis1 robustly interacts with Mfns and OPA1, while the interaction between hFis1 and Drp1 is weak and transient, and is detectable only after chemical crosslinking, in agreement with a previous report (Yoon *et al*, 2003). Furthermore, knockdown of either OPA1 (Fig 3G) or Mfn1/2 (Fig 3H) using siRNAs in WT 293T cells did not affect the binding of hFis1 to respectively Mfn1/Mfn2 or OPA1, indicating that the binding of hFis1 to Mfn1/2 and to OPA1 were independent events. Likewise, knockdown of hFis1 by siRNA did not affect the interaction of

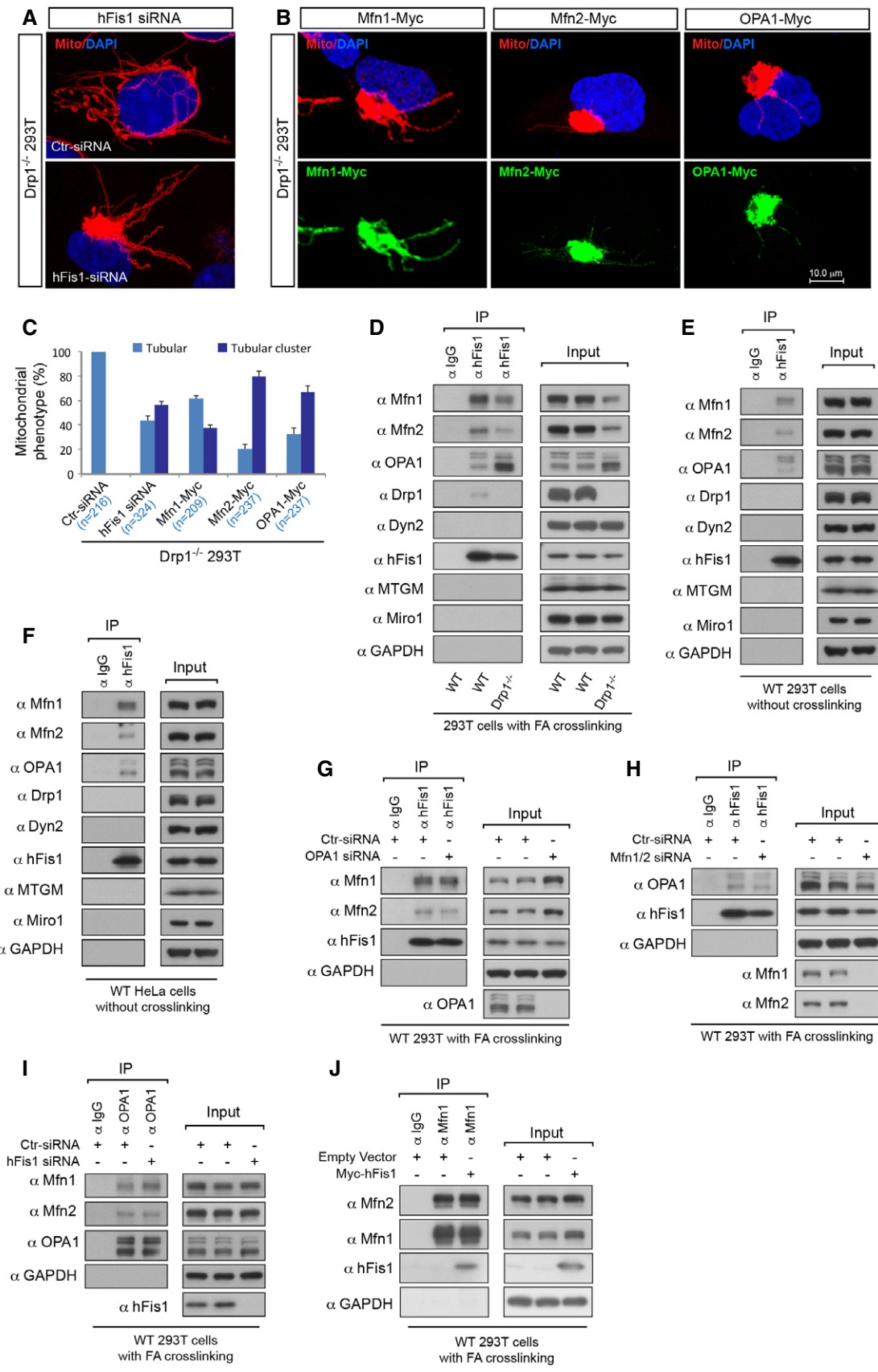

Figure 3.

◄

**Figure 3.  Mfn1, Mfn2, and OPA1 but not Drp1 and Dyn2 are major hFis1-binding partners.**

A    Confocal images of mitochondrial morphology in Drp1$^{-/-}$ 293T cells transfected with scrambled siRNA (control) or with hFis1-siRNA and then stained with MitoTracker (red).

B    Confocal images of mitochondrial morphology in Drp1$^{-/-}$ 293T cells transfected with either Mfn1-Myc, Mfn2-Myc, or OPA1-Myc, stained with MitoTracker (red) followed by immunostaining with anti-Myc antibody (green).

C    Percentages (mean ± SEM) of cells with indicated mitochondrial morphologies in Drp1$^{-/-}$ 293T cells transfected with either scrambled siRNA (Ctr), hFis1 siRNA, Mfn1-Myc, Mfn2-Myc, or OPA1-Myc in three independent experiments for each condition (*n* represents the number of cells analyzed).

D    hFis1 interacts with Mfn1, Mfn2, and OPA1 as well as Drp1, but not with Dyn2 at endogenous levels following chemical crosslinking. Wild-type (WT) and Drp1$^{-/-}$ 293T cells were *in vivo* crosslinked with 1% formaldehyde (FA), and cell lysates were used for co-immunoprecipitation (IP) with Protein G beads bound to rabbit normal IgG (negative control) or rabbit anti-hFis1 antibody as indicated, followed by immunoblotting with indicated antibodies.

E, F    hFis1 binds to Mfn1, Mfn2, and OPA1 at endogenous levels also in the absence of chemical crosslinking. Cell lysates prepared from WT 293T (E) and HeLa (F) cells without chemical crosslinking were used for co-immunoprecipitation (IP) with Protein G beads bound to rabbit normal IgG (negative control) or rabbit anti-hFis1 antibody as indicated, followed by Western blotting with indicated antibodies.

G, H    Interaction of hFis1 with Mfn1/2 and with OPA1 are independent events. WT 293T cells were treated with control, OPA1 (G), or Mfn1 plus Mfn2 (H) siRNA, followed by *in vivo* crosslinking with 1% FA. Cell lysates were used for co-IP with Protein G beads bound to rabbit normal IgG (negative control) or rabbit anti-hFis1 antibody as indicated, followed by immunoblotting with indicated antibodies.

I    Interactions between Mfn1/2 and OPA1 occur independent of hFis1. WT 293T cells were treated with control or hFis1 siRNA, followed by *in vivo* crosslinking with 1% FA. Cell lysates were used for co-IP with Protein G beads bound to mouse normal IgG (negative control) or mouse anti-OPA1 antibody as indicated, followed by immunoblotting with indicated antibodies.

J    Interaction between Mfn1 and Mfn2 is not affected by hFis1 overexpression. 293T cells were transfected with empty vector or Myc-hFis1, followed by *in vivo* crosslinking with 1% FA. Cell lysates were used for co-IP with Protein G beads bound to mouse normal IgG or mouse anti-Mfn1 antibody, followed by immunoblotting with indicated antibodies.

Source data are available online for this figure.

OPA1 with Mfn1 and Mfn2 (Fig 3I), and overexpression of Myc-hFis1 did not affect the endogenous interaction between Mfn1 and Mfn2 (Fig 3J). In conclusion, these data suggest that Mfn1/2 and OPA1, but not the pro-fission proteins Drp1 and Dyn2, are major hFis1-binding GTPase partners.

We next set out to establish which region in hFis1 is responsible for the interactions with Mfns and OPA1. hFis1 is anchored to the MOM through a single C-terminal TM domain with the bulk of the protein including a tetratricopeptide repeat (TPR) domain (consisting of TPR1 and TPR2) facing the cytosol (Fig 4A) (James *et al*, 2003; Yoon *et al*, 2003; Suzuki *et al*, 2005). To define the hFis1 region responsible for the interactions with Mfns and OPA1, several N- or C-terminally truncated, Myc-tagged hFis1 mutants were expressed in hFis1$^{-/-}$ 293T cells generated by CRISPR/Cas9-mediated gene editing (Appendix Fig S3). The interaction of these hFis1 mutants with Mfn1, Mfn2, and OPA1 was evaluated by co-IP, and the results are summarized in Fig 4B, left panel and C. Like full-length hFis1, hFis1$^{\Delta 1-31}$, and hFis1$^{\Delta 1-60}$ interacted with the three pro-fusion GTPases, indicating that the first N-terminal 60 residues of hFis1 are not required for the interactions. In addition, we noticed that hFis1 lacking the first 31 residues (hFis1$^{\Delta 1-31}$) showed an increased interaction with the pro-fusion GTPases. In contrast, the cytosolic mutant Myc-hFis1$^{\Delta TM/C}$ had lost the ability to interact with Mfn1, Mfn2, or OPA1, which indicates that mitochondrial localization is required for hFis1's interaction with the pro-fusion GTPases. Due to rapid degradation of the truncated mutant Myc-hFis1$^{\Delta 1-90}$ (lacking the first 90 residues), we performed co-IP using GFP-fused N-terminally to the truncated hFis1 mutants, i.e., GFP-hFis1$^{\Delta 1-90}$ (lacking the first 90 residues) and GFP-hFis1$^{\Delta 1-121}$ (only containing the TM domain and C-terminal tail) to further define the region of hFis1 responsible for interaction with Mfns and OPA1. We found that GFP-hFis1$^{\Delta 1-90}$ and GFP-hFis1$^{\Delta 1-121}$ retained interaction with Mfns and OPA1 although the interactions were much weaker compared to the full-length GFP-hFis1 (Fig 4B, right panel and C). Together, these findings indicate that part of the cytosolic portion of hFis1, including the TPR2, TM domain, and C-terminal tail, are

required for interaction with Mfns and OPA1, whereas the N-terminal portion, including TPR1, is dispensable.

## hFis1 negatively regulates mitochondrial fusion

The robust interaction with the pro-fusion GTPases (Mfns and OPA1) suggests that hFis1 may play a role in regulating mitochondrial fusion. To test this, we first evaluated the effects of hFis1 overexpression and knockdown on the extent of mitochondrial fusion in WT 293T cells using a polyethylene glycol (PEG)-induced cell fusion assay (Chen *et al*, 2003; Zhao *et al*, 2011). Two WT 293T cell lines stably expressing mitochondria-targeting green or red fluorescent proteins (mitoGFP or mitoRFP) were generated. MitoGFP- and mitoRFP-labeled 293T cells were mixed and transfected with Myc-hFis1 plasmid or with empty vector as control. Cycloheximide was added 30 min before cell fusion to inhibit synthesis of the fluorescent protein. Five hours after cell fusion, hybrid polykaryons were observed by confocal microscopy, and mitochondrial fusion was determined by the mixing of green and red fluorescence signals (yellow) in the polykaryons. The extent of mitochondrial fusion, i.e., co-localization between green and red fluorescence signals, in each polykaryon containing two nuclei was further analyzed by the Pearson's correlation coefficient (PCC). A relatively high level (PCC, $0.60 \pm 0.03$) of mitochondrial fusion was observed in 293T hybrid cells transfected with empty vector (Fig 5A, left panel and B). In contrast, overexpression of hFis1 resulted in reduced mitochondrial fusion (PCC, $0.26 \pm 0.03$, $P < 0.0001$) in polykaryons, compared to empty vector controls (Fig 5A, right panel and B) and this was accompanied by a fragmented mitochondrial phenotype, suggesting that overexpression of hFis1 might impair mitochondrial fusion.

To corroborate these data, we performed a photoactivatable GFP-based fusion assay (Karbowski *et al*, 2004), in which a mitochondrial matrix-targeted photoactivatable green fluorescent protein (mito-PAGFP) was used to monitor and quantify the alteration of mitochondrial fusion in WT 293T cells transfected with empty vector (control) or Myc-hFis1 at different time points. After

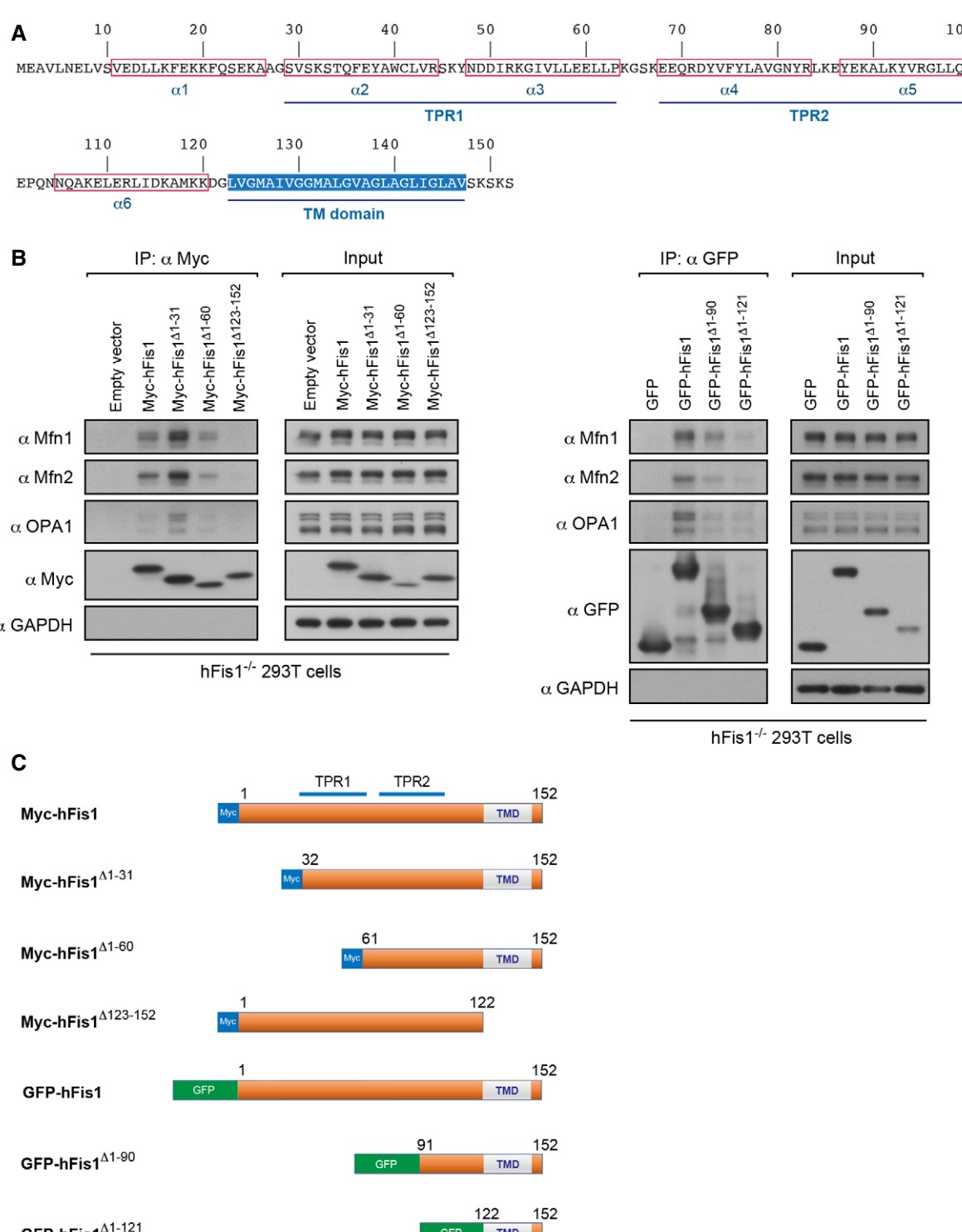

**Figure 4.  Domain requirements in hFis1 for its interaction with Mfns and OPA1.**

A   Amino acid sequence of human Fis1. The location of the six α-helices, the tetratricopeptide repeat (TPR) motifs, and transmembrane domain (TM) are indicated.

B   Interaction of different hFis1 mutants with Mfns and OPA1. hFis1$^{-/-}$ 293T cells were transfected with the full-length and the truncated hFis1 mutants as indicated, followed by *in vivo* crosslinking with 1% FA. Cell lysates were used for co-IP with anti-Myc or anti-GFP agarose beads, and the immunoprecipitates were analyzed by Western blotting with indicated antibodies.

C   Summary of hFis1-mutated constructs used in this experiment.

Source data are available online for this figure.

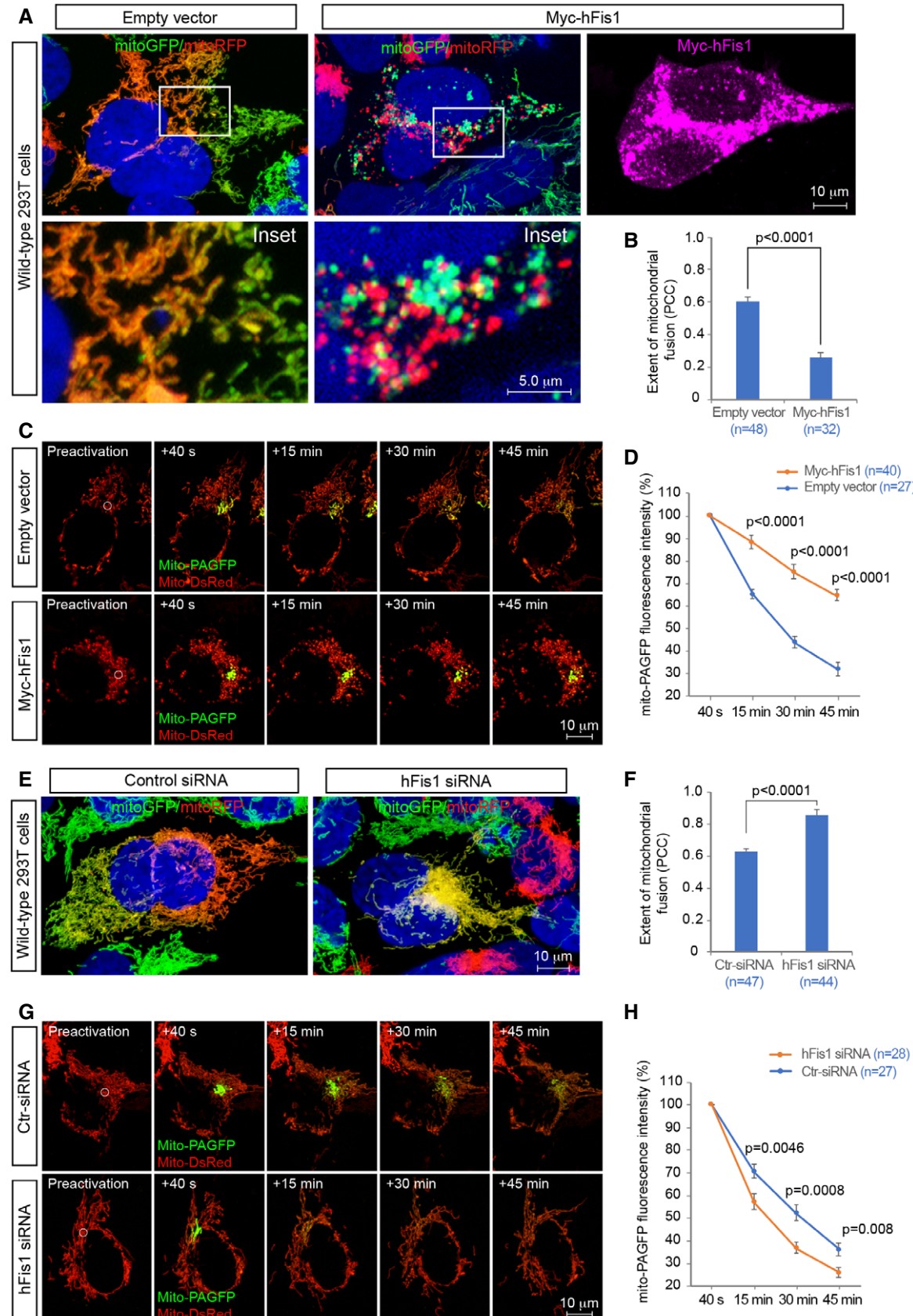

Figure 5.

**Figure 5. hFis1 impairs mitochondrial fusion in wild-type 293T cells.**

A, B   Representative confocal images of mitochondrial fusion in WT 293T polykaryons in cultures subjected to the PEG-based fusion assay in the presence or absence of exogenous hFis1. WT 293T cells stably expressing mitoGFP or mitoRFP were co-cultured, subsequently transfected with empty vector (control) or with Myc-hFis1 as indicated and then treated with PEG for cell fusion. Mitochondrial fusion is indicated by co-localization of mitoGFP and mitoRFP (i.e., yellow mitochondria). Left panel: empty vector transfected; right panel: Myc-hFis1 transfected. Insets represent magnification of the boxed areas in upper panel (A). Quantitative analysis of the extent of mitochondrial fusion in individual hybrid cells was performed by the Pearson's correlation coefficient (PCC) in three independent experiments for each condition and data summarized in (B).

C, D   Representative confocal images of mitochondrial fusion in WT 293T cells assessed by the mito-PAGFP-based fusion assay in the presence or absence of exogenous hFis1. WT 293T cells co-transfected with mito-PAGFP (0.5 µg), mito-DsRed (0.2 µg), and either empty vector (0.5 µg, the upper panel) or hFis1 (0.5 µg, the lower panel) were photoactivated in a small region of interest (ROI) (white circle, 3 µm diameter) in preactivation images of mitochondria seen by mito-DsRed (red). After photoactivation, sequential Z-stack images with photoactivated GFP (green) and mitochondrial marker (mito-DsRed) were collected at indicated time points using series of Z-sections from the top to the cell bottom with intervals between sections set to 0.5–0.75 µm (C). Mitochondrial fusion was quantified by analyzing changes in fluorescence intensity of photoactivated mito-PAGFP in ROIs at 40 s, 15, 30, and 45 min after photoactivation. The dilution rates (percentage) of the GFP fluorescence intensity at different time points were normalized by the fluorescence intensity at 40 s after photoactivation (D).

E, F   Representative confocal images of mitochondrial fusion in WT 293T polykaryons in cultures subjected to the PEG-based fusion assay in the presence or absence of endogenous hFis1. Both WT 293T cells stably expressing mitoGFP or mitoRFP were transfected with scrambled siRNA (control siRNA) or hFis1-siRNA as indicated and then co-cultured and fused using PEG treatment. Left panel: control siRNA transfected; right panel: hFis1-siRNA transfected (E). Quantitative analysis of the mitochondrial fusion in individual hybrid cells was performed by the Pearson's correlation coefficient in three independent experiments and data summarized in (F).

G, H   Representative confocal images of mitochondrial fusion in WT 293T cells assessed by the mito-PAGFP-based fusion assay in the presence or absence of endogenous hFis1. Scrambled siRNA or hFis1 siRNA-treated cells were co-transfected with mito-PAGFP and mito-DsRed and were subsequently photoactivated (G) as described in (C). Mitochondrial fusion was quantified (H) as described in (D).

Data information: Data are expressed as means ± SEM and were statistically analyzed by Student's *t*-test. *n* represents the number of cells analyzed (B, D, F, and H).

photoactivation of a small region of interest (ROI), diffusion and dilution of the photoactivated GFP fluorescence within the mitochondrial network were tracked in a time course experiment. The decrease in GFP fluorescence intensity within the photoactivated ROI at different time points was used for measuring the rates of mitochondrial fusion (Karbowski *et al*, 2014). This assay revealed that overexpression of hFis1 in cells significantly reduced diffusion of the mito-PAGFP signal away from the photoactivated ROI, compared to empty vector control (Fig 5C and D), indicating that hFis1 overexpression reduces the rate of mitochondrial fusion.

Given a potential inhibitory role of hFis1 in mitochondrial fusion, depletion of endogenous hFis1 may be expected to lead to the opposite effect, i.e., increased mitochondrial fusion. We therefore performed the PEG-based and the photoactivatable GFP-based fusion assays after knockdown of hFis1 by siRNA. The efficiency of hFis1 siRNA knockdown in cell fusion assays was confirmed by Western blots (Fig EV3A). Ablation of endogenous hFis1 by siRNA resulted in enhanced mitochondrial fusion in WT 293T polykaryons (PCC, 0.86 ± 0.03) compared to controls transfected with scrambled siRNA (PCC, 0.63 ± 0.01, *P* < 0.0001) (Fig 5E and F) in the PEG-based assay. In line with this, the mito-PAGFP-based assay revealed that knockdown of hFis1 by siRNA resulted in a faster diffusion of mito-PAGFP fluorescence after photoactivation than in scrambled siRNA-treated control cells (Fig 5G and H). These results further support that hFis1 acts as an inhibitor of the mitochondrial fusion machinery in WT 293T cells.

We next evaluated the effect of overexpression and knockdown of hFis1 on mitochondrial fusion in Drp1-deficient 293T cells, using the same experimental setup. Similar to the observations in WT cells, overexpression of hFis1 in Drp1$^{-/-}$ 293T cells reduced the extent of mitochondrial fusion (PCC, 0.13 ± 0.01) compared to control cells transfected with empty vector (PCC, 0.54 ± 0.02, *P* < 0.0001) in polykaryons (Fig 6A and B). Likewise, overexpression of hFis1 greatly reduced dispersion of the mito-PAGFP fluorescence away from the photoactivated ROIs compared to controls (Fig 6C and D). Conversely, knockdown of hFis1 by siRNA (Fig EV3B) in Drp1$^{-/-}$ cells significantly enhanced the extent of

mitochondrial fusion (PCC, 0.83 ± 0.02) compared to controls (PCC, 0.59 ± 0.03, *P* < 0.0001) in polykaryons (Fig 6E and F) and also led to a rapid dispersion of the mito-PAGFP fluorescence away from the photoactivated ROIs compared to controls (Fig 6G and H). In summary, these findings support the notion that hFis1 acts as an inhibitor of mitochondrial fusion and impedes mitochondrial fusion regardless of whether Drp1 is present or not in the cells.

### hFis1 impairs the activity of pro-fusion but not pro-fission GTPases

The interaction of hFis1 with Mfn1, Mfn2, and OPA1 prompted us to investigate whether the effect of hFis1 on mitochondrial fusion was caused by inhibition of the GTPase activity in these pro-fusion proteins. To test this, we performed an *in vitro* GTPase activity assay as previously described (Stafa *et al*, 2012; Wang *et al*, 2012; Samant *et al*, 2014). Myc-tagged Mfn1, Mfn2, and OPA1 were transiently expressed in WT 293T cells and immunopurified by anti-Myc agarose beads, followed by measurement of the GTPase activity after incubation with or without recombinant hFis1 protein. We found that the presence of hFis1 protein reduced the enzymatic activity of Mfn1 and Mfn2, as well as OPA1 (Fig 7A). The mutant Mfn2$^{K109A}$ showed reduced GTPase activity compared to WT Mfn2 and the addition of hFis1 further reduced the GTPase activity of Mfn2$^{K109A}$ (Fig 7A). Immunopurified Mfn1-Myc, Mfn2-Myc, Mfn2$^{K109A}$-Myc, and OPA1-Myc as well as recombinant GST-hFis1 and GST (negative control) proteins used in the experiments shown in Fig 7A were confirmed by Western blot analysis (Fig 7B).

We next assessed the effect of hFis1 on the GTPase activity of the pro-fission GTPase Drp1. To exclude potential interference of endogenous Drp1, we expressed untagged Drp1 and Drp1$^{Q34A}$ respectively in Drp1$^{-/-}$ cells followed by co-IP with anti-Drp1 antibody. The addition of hFis1 in the GTPase assay did not affect the enzymatic activity of Drp1 (Fig 7C), consistent with a previous study (Yoon *et al*, 2003). In contrast, the mutant Drp1$^{Q34A}$ showed greatly reduced GTPase activity (Fig 7C), in line with a previous report (Wenger *et al*, 2013). Similarly, the GTPase activity of

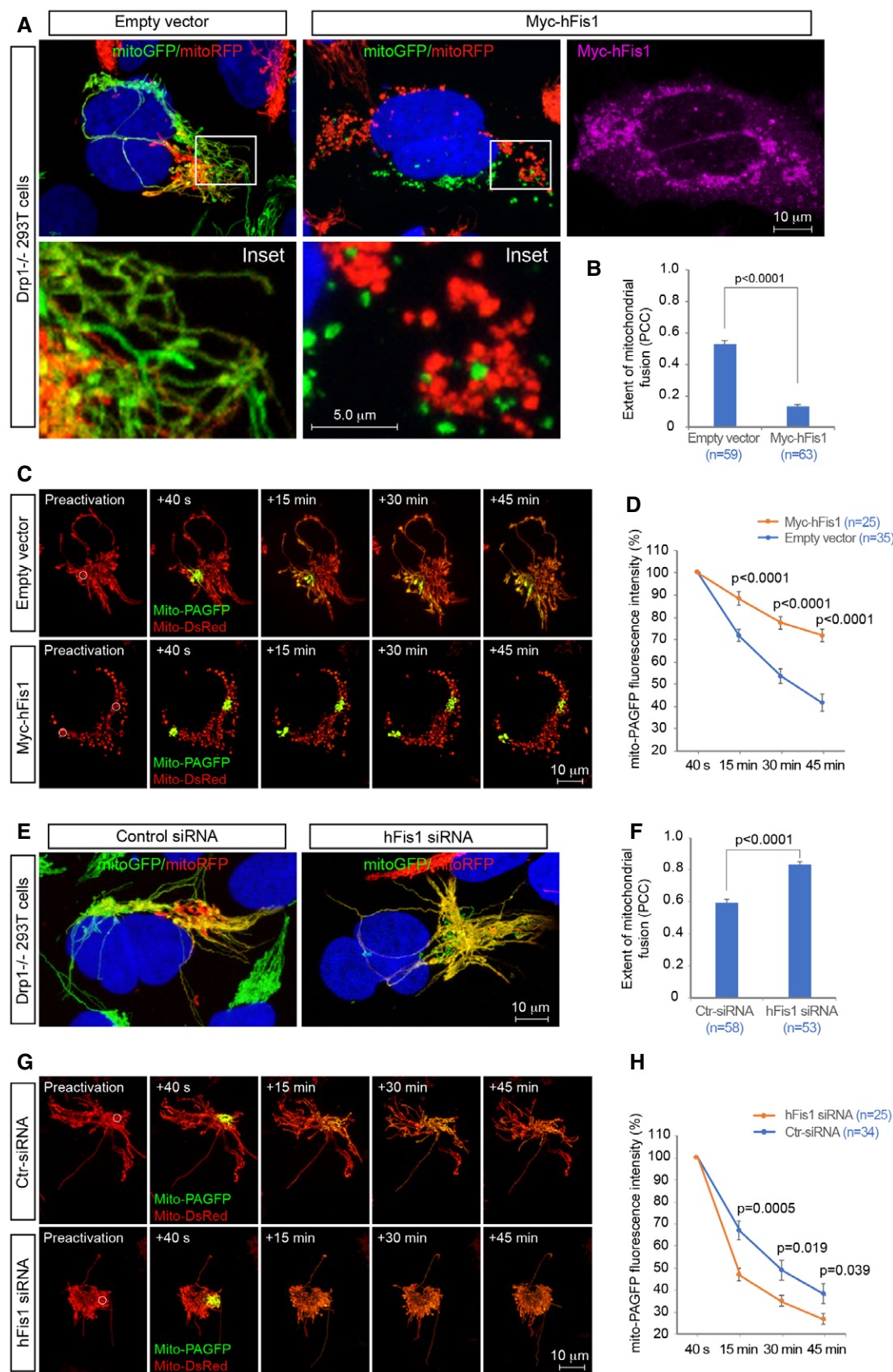

**Figure 6.**

**Figure 6.  hFis1 impairs mitochondrial fusion in Drp1-deficient 293T cells.**

A, B    Representative confocal images of mitochondrial fusion in Drp1$^{-/-}$ 293T polykaryons in cultures subjected to the PEG-based fusion assay in the presence or absence of exogenous hFis1. Drp1$^{-/-}$ cells stably expressing mitoGFP or mitoRFP were co-cultured, subsequently transfected with empty vector (control) or Myc-hFis1 as indicated and then stimulated with PEG treatment for cell fusion, followed by immunostaining with anti-Myc antibody (green). Mitochondrial fusion was indicated by co-localization of mitoGFP and mitoRFP (i.e., yellow mitochondria). Left panel: empty vector transfected; right panel: Myc-hFis1 transfected. Insets represent magnification of the boxed areas in upper panel (A). Quantitative analyses for measuring the extent of mitochondrial fusion in individual hybrid cells were performed by the Pearson's correlation coefficient (PCC) in three independent experiments and summarized in (B).

C, D    Representative confocal images of mitochondrial fusion in Drp1$^{-/-}$ 293T cells assessed by the mito-PAGFP-based fusion assay in the presence or absence of exogenous hFis1. Drp1$^{-/-}$ cells co-transfected with mito-PAGFP, mito-DsRed, and either empty vector (the upper panel) or hFis1 (the lower panel) were photoactivated in a small region of interest (ROI) (white circle, 3 μm diameter) as indicated in preactivation images of mitochondria seen by mito-DsRed (red). After photoactivation, Z-stack images with photoactivated GFP (green) and mitochondrial marker (mito-DsRed) were collected at times 40 s, 15, 30, and 45 min as indicated (C). Mitochondrial fusion was quantified by analysis of changes in the fluorescence intensity of photoactivated mito-PAGFP in ROIs at 40 s, 15, 30, and 45 min after photoactivation. The dilution rates (percentage) of the GFP fluorescence intensity at different time points were normalized by the fluorescence intensity at 40 s after photoactivation (D).

E, F    Representative confocal images of mitochondrial fusion in Drp1$^{-/-}$ 293T polykaryons in cultures subjected to the PEG-based fusion assay in the presence or absence of endogenous hFis1. Drp1$^{-/-}$ cells stably expressing mitoGFP or mitoRFP were transfected with scrambled siRNA (control siRNA) or with hFis1-siRNA as indicated and then co-cultured and fused by PEG treatment. Left panel: control siRNA transfected; right panel: hFis1-siRNA transfected (E). Quantitative analysis of extent of mitochondrial fusion in individual hybrid cells was performed by the Pearson's correlation coefficient (PCC) in three independent experiments and summarized in (F).

G, H    Representative confocal images of mitochondrial fusion in Drp1$^{-/-}$ 293T cells monitored by the mito-PAGFP-based fusion assay in the presence or absence of endogenous hFis1. Scrambled siRNA or hFis1 siRNA-treated cells were co-transfected with mito-PAGFP and mito-DsRed and were subsequently photoactivated (G) as described in (C). Mitochondrial fusion was quantified by analysis of changes in the fluorescence intensity of photoactivated mito-PAGFP (H) as described in (D).

Data information: Data are expressed as means ± SEM and were statistically analyzed by Student's t-test. n represents the number of cells analyzed (B, D, F, and H).

endogenous Dyn2 was not affected by recombinant hFis1 (Fig 7E). Immunopurified Drp1 and Dyn2 were confirmed by Western blot (Fig 7D and F). These results indicate that hFis1 mainly regulates the GTPase activity of Mfn1, Mfn2, and OPA1, but not of Drp1 and Dyn2.

We next tested whether co-expression of hFis1 with Mfn1, Mfn2, or OPA1 affected the hFis1-induced mitochondrial fragmentation in Drp1-deficient cells. Compared with expression of hFis1 alone, co-expression with pro-fusion GTPase proteins (Mfn1, Mfn2 or OPA1) more or less reduced the hFis1-mediated fragmentation; especially co-expression with Mfn2 strongly reversed the mitochondrial fragmentation induced by hFis1 overexpression in Drp1$^{-/-}$ cells (Figs 7G and EV4). These data further support the notion that the pro-fusion GTPases are involved in regulating hFis1-induced fragmentation.

### Disruption of the fusion machinery in Drp1-deficient cells phenocopies the mitochondrial fragmentation induced by hFis1 overexpression

As hFis1 inhibited the GTPase activity of Mfn1, Mfn2, and OPA1, we were interested in further exploring whether destruction of the fusion machinery would mimic the mitochondrial fragmentation phenotype induced by hFis1 overexpression in Drp1-deficient cells. To address this, we first knocked out the pro-fusion GTPase OPA1 in Drp1$^{-/-}$ 293T cells using CRISPR/Cas9-mediated gene editing, generating a Drp1$^{-/-}$/OPA1$^{-/-}$ double knockout (Drp1/OPA1$^{DKO}$) 293T cell line (Fig 8A and B, and Appendix Fig S4). Ablation of OPA1 in the Drp1$^{-/-}$ cells partially reversed the super-fused mitochondrial network induced by loss of Drp1, resulting in 19.2 ± 1.7% of cells with fragmented mitochondria (Fig 8B and D), and accompanied by an asymmetrical perinuclear aggregation of fragmented (Aggr/frag) or bulb-like/tubular (Bulb/tubular) mitochondria predominantly localized to one side of the nucleus (Fig 8B, right panel).

Knockdown of Mfn2 by siRNA in the Drp1/OPA1$^{DKO}$ cells did not result in any significant difference in mitochondrial morphology

as compared to Drp1/OPA1$^{DKO}$ cells treated with control (scrambled) siRNA (Fig 8C, lower panel and D). In contrast, knockdown of Mfn1 by siRNA in Drp1/OPA1$^{DKO}$ cells triggered extensive mitochondrial fragmentation in most of the cells (81.8 ± 1.9%) and double knockdown of Mfn1 and Mfn2 resulted in a further increase in cells (98.5 ± 0.4%) with fragmented mitochondria, compared to the control siRNA-treated group (19.2 ± 1.7%) (Fig 8C, lower panel and D). The extent of mitochondrial fragmentation in Drp1/OPA1$^{DKO}$ cells depleted of Mfn1 or Mfn1/2 was comparable to the effect of hFis1 overexpression in Drp1$^{-/-}$ cells (90 ± 2.6% of cells with fragmented mitochondria) (Fig 8C, upper panel and D). Additionally, we noticed that depletion of Mfn1 in Drp1/OPA1$^{DKO}$ cells led to that the asymmetrical perinuclear clustering of mitochondria observed in Drp1/OPA1$^{DKO}$ cells (Fig 8B, right panel) reverted to a more evenly dispersed pattern of fragmented mitochondria throughout the cell. Furthermore, overexpression of hFis1 in the Drp1/OPA1$^{DKO}$ cells still induced extensive mitochondrial fragmentation (98.8 ± 2.7% of cells with fragmented mitochondria), similar to that seen in Drp1/OPA1$^{DKO}$ cells depleted of both Mfn1 and Mfn2 (Fig 8C, lower panel and D).

Finally, we evaluated the potential role of Dyn2 and found that depletion of Dyn2 significantly increased the number of Drp1/OPA1$^{DKO}$ cells with tubular mitochondria (96.1% ± 0.49) compared to the control siRNA group (80.8 ± 1.7%) (Fig 8C and D, $P < 0.0001$) and also partially decreased mitochondrial fragmentation induced by knockdown of Mfn1 in Drp1/OPA1$^{DKO}$ cells (71.8 ± 2.6%) compared to the control group (81.8 ± 1.9%) treated with Mfn1 siRNA alone (Fig 8C and D, $P < 0.001$). This suggests that Dyn2 is required for mitochondrial division that occurs when the canonical fission and fusion machineries are simultaneously blocked. However, depletion of Dyn2 in Drp1/OPA1$^{DKO}$ cells did not prevent fragmentation induced by overexpression of hFis1 (Fig 8C and D). Knockdown of the indicated proteins by siRNA was confirmed by Western blotting (Fig 8E). In sum, the data indicate that destruction of the mitochondrial fusion machinery in Drp1-deficient cells phenocopies hFis1 overexpression-induced mitochondrial

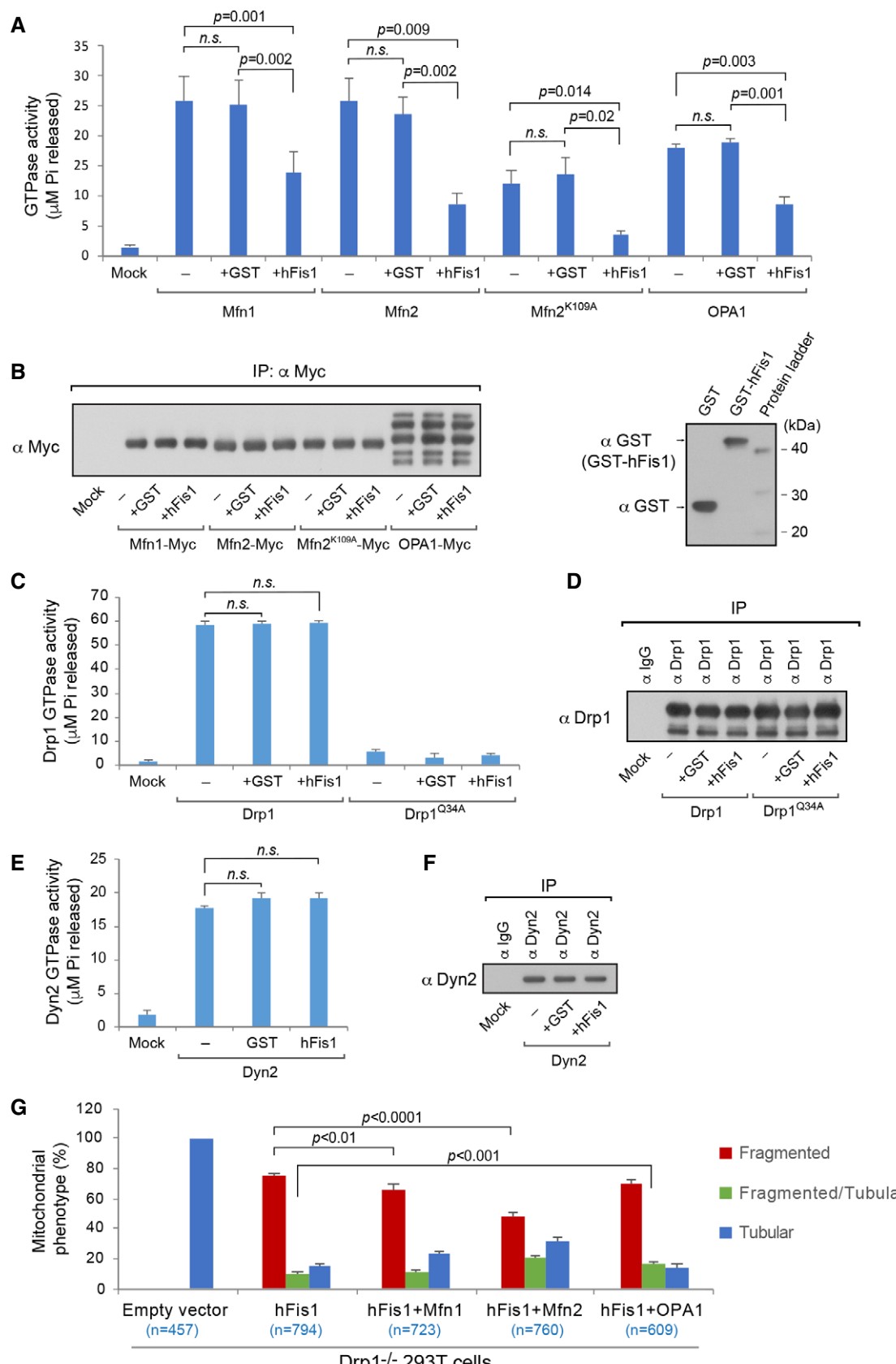

Figure 7.

◄

**Figure 7.  Effects of hFis1 on the GTPase activity of pro-fusion proteins Mfn1, Mfn2, and OPA1 as well as pro-fission proteins Drp1 and Dyn2.**

A, B  GTP hydrolysis activities of Mfn1, Mfn2, Mfn2[K109A], and OPA1 in the presence or absence of recombinant GST-hFis1 or GST (negative control) were determined by measuring the concentration of free phosphate (P$_i$) released from GTP. Myc-tagged Mfn1, Mfn2, Mfn2[K109A], and OPA1 were transiently expressed separately in 293T cells and immunopurified by anti-Myc agarose beads. GTPase activities of immunopurified Myc-tagged Mfn1, Mfn2, Mfn2[K109A], and OPA1 were determined after treatment with or without recombinant GST-hFis1 or GST protein (A). The input levels of immunopurified Mfn1-Myc, Mfn2-Myc, and OPA1-Myc (left panel) as well as recombinant GST-hFis1 (~42 kDa) and GST (~26 kDa) (right panel) were assessed by immunoblotting with indicated antibodies (B).

C, D  GTP hydrolysis activities of Drp1 in the presence or absence of recombinant GST-hFis1 or GST (negative control) were determined as described in (A). Untagged WT Drp1 and Drp1[Q34A] were transiently expressed in Drp1$^{-/-}$ 293T cells and immunopurified by protein G beads pre-incubated with Drp1 antibody. GTPase activities of immunopurified Drp1 and Drp1[Q34A] were determined after treatment with or without recombinant hFis1 or GST protein (C). The input levels of immunopurified Drp1 and Drp1[Q34A] were assessed by immunoblotting with anti-Drp1 antibody (D).

E, F  GTP hydrolysis activities of Dyn2 in the presence or absence of recombinant GST-hFis1 or GST (negative control) were determined as described in (A). Endogenous Dyn2 in Drp1$^{-/-}$ 293T cells was immunopurified by protein G beads pre-incubated with anti-Dyn2 antibody. GTPase activity of immunopurified Dyn2 was determined after treatment with or without recombinant GST-hFis1 or GST protein (E). The input levels of immunopurified Dyn2 were assessed by immunoblotting with anti-Dyn2 antibody (F).

G  Summary of the effects of pro-fusion GTPases (Mfns and OPA1) on hFis1-induced mitochondrial fragmentation in Drp1$^{-/-}$ cells. Percentages (mean ± SEM) of cells with indicated mitochondrial morphologies in Drp1$^{-/-}$ 293T cells co-expressing Myc-hFis1 and either empty vector, Mfn1-Myc, Mfn2-Myc, or OPA1-Myc in three independent experiments. *n* represents the number of cells analyzed. Corresponding confocal images shown in Fig EV4.

Data information: Data are expressed as means ± SEM (*n* = 4 experiments in each condition) and were statistically analyzed by Student's *t*-test (A, C and E). n.s., not significant.

Source data are available online for this figure.

fragmentation, and that Dyn2 is to some extent involved in this process. However, Dyn2 was not required for the hFis1-induced fragmentation observed in Drp1/OPA1$^{DKO}$ cells.

## Inhibition of F-actin prevents the mitochondrial fragmentation induced by hFis1 overexpression in Drp1-deficient cells

The actin cytoskeleton has recently emerged as an important player in mitochondrial dynamics, and actin depolymerization has been reported to impair mitochondrial fission by inhibiting Drp1 recruitment to mitochondria (De Vos *et al*, 2005; Ji *et al*, 2015; Li *et al*, 2015; Manor *et al*, 2015; Hatch *et al*, 2016; Moore *et al*, 2016). We were interested in learning whether the actin filaments (F-actin) were important also for the pro-fission effects of hFis1. To test this, latrunculin B (LatB; a drug that depolymerizes F-actin) was added to Myc-hFis1 expressing Drp1-deficient cells. Interestingly, LatB treatment significantly prevented mitochondrial fragmentation in Myc-hFis1 expressing Drp1$^{-/-}$ cells (55.4 ± 2.7%) compared to control cells (74.7 ± 1.9%, *P* < 0.0001) (Fig EV5A and B), indicating that the actin cytoskeleton is involved in hFis1-induced fragmentation also in the absence of Drp1.

To further explore whether LatB could reverse hFis1-induced mitochondrial fragmentation in Drp1$^{-/-}$ cells, Myc-hFis1 was transiently transfected in three sets of Drp1$^{-/-}$ cells and incubated for 20 h (Fig EV5C and D). One group of cells (Group A) was harvested directly after the 20-h transfection period without LatB treatment, whereas Group B and Group C were treated with LatB and DMSO (control) respectively for an additional 16 h thereafter. When cultures were treated with LatB for 16 h (Group B) after transfection of Myc-hFis1 for 20 h, we found that this did not reduce the number of cells with fragmented mitochondria (93.7 ± 0.8%) compared to Group A (85.3 ± 1.1%), but decreased mitochondrial fragmentation compared to the DMSO control (96.1 ± 0.5%) incubated in parallel for an additional 16 h (Group C) (Fig EV5C and D). This suggests that LatB treatment efficiently prevented mitochondrial fragmentation but was incapable of reversing it once fragmentation had occurred in Myc-hFis1 overexpressing Drp1$^{-/-}$ cells. These results, taken together with the results from mitochondrial fusion assays performed in both WT and Drp1$^{-/-}$ 293T cells exogenously

expressing Myc-hFis1 (see Figs 5 and 6), support the notion that the fragmented mitochondria in Myc-hFis1 expressing cells are fusion incompetent, in line with a previous report (Yoon *et al*, 2003). We suggest that under these conditions mitochondrial fission occurs via a Drp1-independent pathway that partially involves actin cytoskeleton.

## Discussion

Central to the control of mitochondrial dynamics is a family of dynamin-related GTPases, which can be divided into the fission-promoting Drp1 and Dyn2 and the fusion-promoting Mfn1, Mfn2, and OPA1. A dynamic regulation of mitochondrial shape is essential for eukaryotic cells in response to various physiological challenges, and it is therefore important to understand the molecular basis for control of mitochondrial dynamics. In addition to the GTPases controlling fusion and fission, another highly evolutionarily conserved protein, Fis1, is an important regulator of mitochondrial dynamics. The yeast homolog Fis1p serves as an essential mitochondrial receptor for recruitment of Drp1 (Dnm1p) to the mitochondrial outer membrane through one of the adaptors Mdv1p and Caf4p (Mozdy *et al*, 2000; Tieu & Nunnari, 2000; Griffin *et al*, 2005; Okamoto & Shaw, 2005; Hoppins *et al*, 2007). In the light of its evolutionary conservation, several studies have implicated Fis1 as a receptor for Drp1 recruitment to mitochondria also in mammalian cells (James *et al*, 2003; Yoon *et al*, 2003), but there are also reports indicating that Fis1 is largely dispensable for Drp1 recruitment and Drp1-mediated mitochondrial fission (Lee *et al*, 2004; Stojanovski *et al*, 2004; Otera *et al*, 2010; Koirala *et al*, 2013; Loson, 2013; Osellame *et al*, 2016).

In this report, we present data indicating that hFis1 promotes fission by acting as a negative regulator of the pro-fusion machinery. Several lines of evidence support this notion. Firstly, hFis1 exerts a fission-promoting effect also in the absence of Drp1 and Dyn2, indicating that these two proteins are largely dispensable for hFis1 function. This observation is in keeping with a previous report, demonstrating that inhibition of Drp1 function only partially prevented mitochondrial fragmentation induced by hFis1

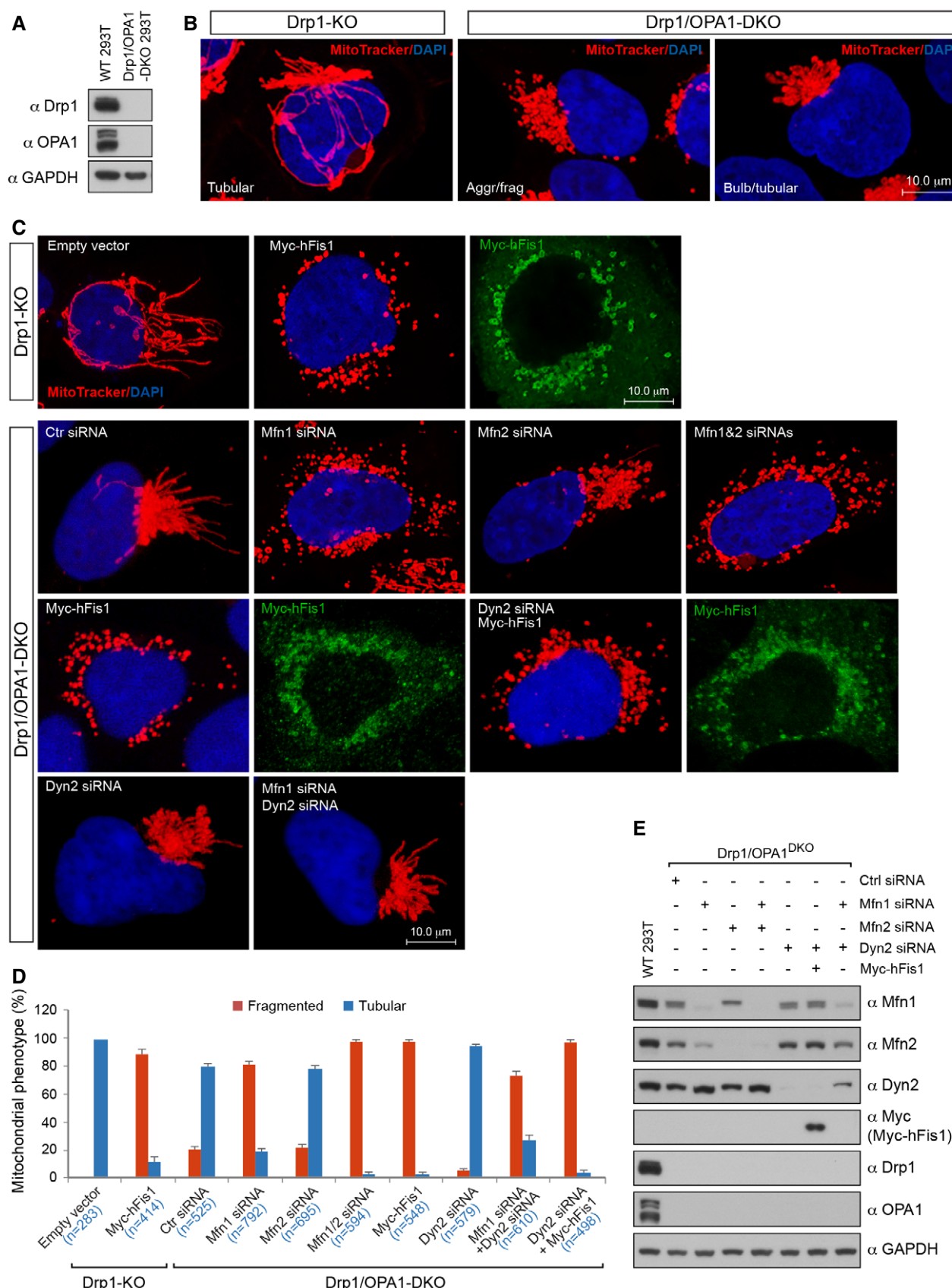

**Figure 8.**

◀

**Figure 8.  Disruption of the fusion machinery in Drp1$^{-/-}$ cells phenocopies the mitochondrial fragmentation induced by hFis1 overexpression.**

A   Ablation of OPA1 using CRISPR-Cas9 gene editing in Drp1$^{-/-}$ 293T cells was confirmed by Western blotting analysis.

B   Confocal images of mitochondrial morphology in Drp1$^{-/-}$ (left panel) and Drp1/OPA1$^{DKO}$ 293T (right panel) cells (Aggr/frag: aggregation/fragmentation of mitochondria; Bulb/tubular: bulb-like/tubular mitochondria).

C   Confocal images of mitochondrial morphology in Drp1/OPA1$^{DKO}$ 293T cells treated with the indicated siRNAs, combinations of siRNAs or with Myc-hFis1 plasmid (lower three panels) compared to mitochondrial morphology in Drp1$^{-/-}$ 293T cells overexpressing Myc-hFis1 or not (upper panel).

D   Percentages (mean ± SEM) of cells with indicated mitochondrial morphologies in Drp1$^{-/-}$ 293T cells transfected with empty vector or Myc-hFis1 plasmid, and in Drp1/OPA1$^{DKO}$ 293T cells treated with the indicated siRNA, combinations of siRNA or Myc-hFis1 plasmid. The data were collected from three independent experiments. Total cell numbers (n) used for statistical analysis are indicated for each condition.

E   Knockdown of the indicated proteins by siRNA in Drp1/OPA1$^{DKO}$ 293T cells was confirmed by Western blotting analysis.

Source data are available online for this figure.

overexpression (Yoon *et al*, 2003). Secondly, we show that hFis1 does not bind Dyn2 and exhibits only a weak interaction with Drp1 under chemical crosslinking conditions, corroborating the previous study (Yoon *et al*, 2003). In contrast, we find that hFis1 interacts more robustly with Mfn1, Mfn2, and OPA1 at endogenous levels, even without crosslinking. Along these lines, hFis1 inhibits the GTPase activity of Mfn1 and Mfn2 as well as OPA1. How Fis1 affects the activity of OPA1 is still enigmatic, but it is plausible that the effect is more indirect, involving proteins located both in the MOM and in the MIM. We observed that the GTPase activity of Drp1 and Dyn2 is not affected by hFis1, in line with a previous report that hFis1 did not change the GTPase activity of Drp1 (Yoon *et al*, 2003). Thirdly, overexpression of hFis1 severely reduces the extent of mitochondrial fusion as determined by the traditional PEG-mediated fusion assay, whereas depletion of hFis1 largely enhances the extent of mitochondrial fusion. Similarly, results from the photoactivatable GFP (mito-PAGFP)-based fusion assay support the notion that hFis1 functions as an inhibitor of the fusion machinery regardless of whether Drp1 is present or not in the cells. Finally, disruption of the fusion machinery phenocopies the mitochondrial fragmentation induced by hFis1 overexpression in Drp1-deficient cells.

The discovery of Drp1- and Dyn2-independent promotion of fission does not diminish the importance of Drp1 and Dyn2; these proteins are important players for the abscission of mitochondria, and recruitment of Drp1 to the MOM is a key step in the fission process (Otera *et al*, 2013; Kraus & Ryan, 2017; Ramachandran, 2018). It was, however, unexpected to learn that the Drp1 recruitment process as well as the role of Fis1 has evolved to become different in yeast and mammalian cells. In yeast, Fis1p is pivotal for recruiting the Drp1 homolog Dnm1p to the mitochondrial outer membrane through one of the adaptors Mdv1p or Caf4p (Mozdy *et al*, 2000; Tieu & Nunnari, 2000; Griffin *et al*, 2005; Okamoto & Shaw, 2005; Hoppins *et al*, 2007). In mammals, no homologs to Mdv1p and Caf4p have yet been found, and Drp1 recruitment is instead conducted by the outer membrane-bound proteins Mff (found in metazoans but not in yeast) and MIEF1/2 (MiD49/51, found only in vertebrates but not in invertebrates, plants, and yeast) (Gandre-Babbe & van der Bliek, 2008; Otera *et al*, 2010; Palmer *et al*, 2011; Zhao *et al*, 2011). The presence of Mff and MIEF1/2 as recruitment proteins for Drp1 may have alleviated Fis1 from a recruitment role and allowed it to diverge to take on a fusion-blocking role in mammals.

The proposed functional divergence between human hFis1 and yeast Fis1p might be related to their differences in protein structure and sequence. Although hFis1 and Fis1p are structurally similar, for instance, they both are anchored in the MOM by a C-terminal TM domain (Mozdy *et al*, 2000; James *et al*, 2003) and their cytosolic domains contain six α-helices that form two tandem tetratricopeptide repeat (TPR) motifs (Suzuki *et al*, 2003, 2005), the protein structural studies suggest that the N-terminal region from Met1 to Gln40 of yeast Fis1p is the least similar to that of hFis1, and it is of note that the N-terminal tail of yeast Fis1p, which is absent in human hFis1, is required for the recruitment of Mdv1p to mitochondria in yeast (Suzuki *et al*, 2005). The functional difference proposed here is in agreement with that human hFis1 cannot rescue the mutant phenotype observed in *fis1*-deficient yeast (Stojanovski *et al*, 2004), whereas yeast Fis1p, when expressed exogenously in HeLa cells, fails to stimulate mitochondrial fragmentation although it can be efficiently targeted to mitochondria (Jofuku *et al*, 2005). One important difference in such a setting is of course the absence of yeast adaptors Mdv1p and Caf4p in human cells. It will be interesting to see if the inhibitory effect described here for human Fis1 on mitochondrial fusion is conserved or not in yeast, and if not, which of the relatively subtle structural differences in Fis1 are responsible for its diverged functions in yeast and mammalian cells.

It is becoming increasingly clear that Fis1 can regulate mitochondrial dynamics via distinct pathways in mammals. Firstly, although Fis1 plays a minor role for Drp1 recruitment under many conditions (Otera *et al*, 2010; Loson *et al*, 2013), Fis1 is involved in Drp1-dependent mitochondrial fission by interacting with and recruiting Drp1 to mitochondria (James *et al*, 2003; Yoon *et al*, 2003; Jofuku *et al*, 2005), especially in response to cell stress-triggered mitochondrial fission, such as mitophagy/apoptosis-related fission, or pathophysiology-associated fission (Kaddour-Djebbar *et al*, 2010; Kim *et al*, 2011; Qi *et al*, 2013; Joshi *et al*, 2018a,b). Secondly, emerging evidence shows that Fis1 is also involved in Drp1- and/or Dyn2-independent mitochondrial fission, for instance, depletion of Drp1 and/or Dyn2 only partially reduced hFis1-induced mitochondrial fragmentation as described in the work presented here as well as in a previous study (Yoon *et al*, 2003). In this regard, a recent report has elucidated a new and intriguing role of Fis1 in mitochondrial fission via mitochondria–lysosome contacts, where TBC1D15, a Rab7 GTPase-activating protein, is recruited to mitochondria by hFis1 to drive lysosomal Rab7 GTP hydrolysis, thereby regulating both lysosomal and mitochondrial dynamics (Wong *et al*, 2018). This finding is corroborated by a study showing that expression of Fis1 or co-expression of Fis1 and TBC1D15 induces mitochondrial fragmentation in Drp1-knockout mouse embryonic fibroblasts (MEFs), suggesting that Fis1 and TBC1D15 function independently of Drp1 (Onoue *et al*, 2013). To add to the Drp1-independent functions for hFis1, we here unveil a novel function of hFis1 in mitochondrial dynamics, i.e., hFis1 acts as an inhibitor of the fusion machinery,

shifting the balance of mitochondrial dynamics toward fission that may favor mitochondrial fission. Together, these data indicate that Fis1 can regulate mitochondrial dynamics via different mechanisms directly or indirectly contributing to mitochondrial fragmentation.

Molecular mechanisms mediating mitochondrial fission have also become more complex in mammals. Although Drp1-driven mitochondrial division represents the canonical mitochondrial fission machinery, emerging evidence suggests the existence of Drp1-independent fission mechanisms (Ishihara *et al*, 2009; Roy *et al*, 2016; Yamashita *et al*, 2016), and additional GTPases, for example, Dyn2 and lysosome-associated Rab7, have been reported to be involved in mitochondrial fission in mammals (Lee *et al*, 2016; Wong *et al*, 2018). Dyn2, like Drp1, is a member of the dynamin family of GTPases and has been proposed to catalyze the final fission event downstream of Drp1-driven mitochondrial division (Lee *et al*, 2016). Interestingly, we show here that knockdown of Dyn2 partially inhibits mitochondrial fission in cells where Drp1 and the pro-fusion Mfn1 and OPA1 were simultaneously depleted, suggesting that Dyn2 can also drive mitochondrial division independently of Drp1, although more work is required to elucidate the underlying mechanisms. In addition, endoplasmic reticulum (ER)–mitochondria contacts, mitochondria–lysosome contacts, and the actin cytoskeleton have been described to be involved in mitochondrial fission (De Vos *et al*, 2005; Friedman *et al*, 2011; Korobova *et al*, 2013; Wong *et al*, 2018). For instance, the ER-mitochondria contacts and the F-actin cytoskeleton are involved in mitochondrial fission induced by infection with *Listeria monocytogenes* in a Drp1-independent manner (Stavru *et al*, 2013). Here, we show that inhibition of F-actin by LatB treatment significantly prevents hFis1-induced mitochondrial fragmentation in Drp1$^{-/-}$ cells, indicating that the actin cytoskeleton is indeed involved in a Drp1-independent fission pathway in hFis1-mediated fusion incompetent cells. In addition, a recent study reports that mitochondrial fragmentation occurs independently of Drp1 during mitophagy (Yamashita *et al*, 2016). Taking these data into consideration, it is not surprising that Fis1-mediated mitochondrial fragmentation can proceed even in the absence of Drp1.

In conclusion, our findings uncover an unanticipated functional difference in Fis1 in the control of mitochondrial dynamics in yeast and mammals and provide a richer and more refined view of how mitochondrial dynamics is controlled in mammalian cells. A future challenge will be to learn in which physiological contexts the different mechanisms are deployed.

# Materials and Methods

### Antibodies and reagents

In this study, the following antibodies were used. Mouse primary antibodies: Myc tag, Drp1, OPA1, cytochrome *c,* and PARP (BD Biosciences); GAPDH (Santa Cruz); Mfn1, Mfn2, and Miro1 (Abcam); GFP and GST (Invitrogen). Rabbit primary antibodies: Dyn2 (Abcam); hFis1 (Atlas Antibodies); Myc tag (Sigma-Aldrich); LC3B (Cell Signaling); and MTGM (Romo1) (Zhao *et al*, 2009). Secondary antibodies: DyLight 488-, 594-, and 649-conjugated anti-mouse and anti-rabbit IgG antibodies (Vector Laboratories) for immunofluorescence and peroxidase-conjugated anti-mouse and anti-rabbit IgG antibodies (GE healthcare) for Western blot. Alexa

Fluor™ 488 Phalloidin (Invitrogen) was used for visualization of F-actin by fluorescence. Dynabeads® Protein G (Invitrogen) and anti-Myc tag antibody agarose (Abcam) and anti-GFP tag antibody agarose (MBL) were used for immunoprecipitation. The Mito-Tracker Red CMXRos (Thermo Fisher) was used for labeling mitochondria. Polyethylene glycol (PEG) 1500 and cycloheximide (Sigma) were used in cell fusion assay. Latrunculin B (LatB) (Sigma-Aldrich) was used for inhibiting actin polymerization as previously described (Wakatsuki *et al*, 2001; Zackroff & Hufnagel, 2002; Korobova *et al*, 2013).

### Cell culture and transfection

293T (HEK293T), HeLa cells, and generated knockout cells were cultured in Dulbecco's modified Eagle's medium (DMEM) (HyClone) with 10% fetal bovine serum (Invitrogen) and 1% PEST antibiotics (Invitrogen) at 37°C in a humidified atmosphere of 5% CO$_2$. Transient transfection of plasmids was performed using Lipofectamine™ 2000 transfection reagent (Invitrogen) according to the manufacturer's protocol. Transfected cells were used for analysis at 18–20 h posttransfection.

Expression plasmids used in this study include Myc-hFis1 (Yu *et al*, 2005); Mfn1-Myc, Mfn2-Myc (Rojo *et al*, 2002); OPA1-Myc (Guillery *et al*, 2008); mitoRFP and mitoGFP (Evrogen Company); mito-PAGFP (Karbowski *et al*, 2004); and mito-DsRed (a kind gift from Dr. Christian Haass, Ludwig-Maximilians-University, Germany). Constructs with hFis1 deletion mutants Myc-hFis1$^{\Delta1–31}$, Myc-hFis1$^{\Delta1–60}$, and Myc-hFis1$^{\Delta TM/C}$ were generated by PCR and cloning into pcDNA3.1 (Invitrogen), and GFP-hFis1, GFP-hFis1$^{\Delta1–90}$, and GFP-hFis1$^{\Delta1–21}$ were cloned into pcDNA3.1/NT-GFP (Invitrogen). Myc-hFis1/yTom5C was constructed in pcDNA3.1 by PCR using synthetic oligonucleotides as described (Jofuku *et al*, 2005). The untagged Drp1 was created by PCR using Myc-Drp1 (Zhu *et al*, 2004) as template and cloned into pcDNA3.1. Site-directed mutagenesis creating Myc-hFis1$^{K149/151A}$, Mfn2$^{K109A}$-Myc, and untagged Drp1$^{Q34A}$ was performed using the QuikChange Lightning Multi Site-Directed Mutagenesis Kit (Agilent). All constructs were verified by sequencing.

### Generation of Drp1$^{-/-}$, hFis1$^{-/-}$, and Drp1/OPA1$^{DKO}$ cell lines

Drp1$^{-/-}$, hFis1$^{-/-}$, and Drp1/OPA1$^{DKO}$ cell lines were established using the CRISPR/Cas9 gene-editing system (Ran *et al*, 2013). The CRISPR/Cas9 vector (Addgene plasmid # 48139) containing a Cas9 nuclease expression cassette and guide RNA cloning cassette was used in this experiment. The Drp1 (Appendix Figs S1A and S2A), hFis1 (Appendix Fig S3A), and OPA1 (Appendix Fig S4A) guide RNAs were designed using the online CHOPCHOP software (https://chopchop.rc.fas.harvard.edu/index.php). The experiments were performed as previously described (Yu *et al*, 2017). Briefly, 293T or HeLa cells were transfected with the CRISPR nuclease vector with the gene-target guide RNA for 24 h, followed by selection in puromycin (3 μg/ml in 293T and 1 μg/ml in HeLa) for 48 h. Subsequently, single cells were picked and cultured in 48-well plates to form colonies. Each colony was validated by Western blotting and PCR cloning followed by DNA sequencing (Eurofins Genomics). The double knockout Drp1/OPA1$^{DKO}$ 293T cell line was generated from Drp1$^{-/-}$ 293T cells.

## Establishment of WT and Drp1$^{-/-}$ 293T cell lines with stable expression of mitoRFP or mitoGFP

The wild-type and Drp1$^{-/-}$ 293T cells were transfected with pTagRFP-mito or pTagGFP-mito plasmid with neomycin (G418) resistance (Evrogen). From the following day until 2 weeks later, the transfected cells were cultured in regular growth medium containing G418 (2.5 mg/ml) for selection of stable expressing colonies. Finally, colonies generated from single cells were picked and cultured. Stable cell lines with expression of either mitoRFP or mitoGFP were validated by fluorescence microscopy.

## RNA interference by siRNA

RNAi experiments were carried out by Lipofectamine™ RNAiMax (Invitrogen) according to the manufacturer's protocol. Briefly, 24 h after initial transfection with siRNA, cells were re-transfected with the same siRNA, incubated for another 48 h and harvested for further investigations. Stealth RNAi™ siRNA Negative Control Kit with similar GC content (Invitrogen) was used as control. For siRNA treatment followed by overexpression of target proteins, cultured cells were first treated with siRNA as described above and transfected with expression plasmids after around 48 h, incubated for another 18 to 20 h, and harvested for further investigation. siRNA oligonucleotides were purchased from Thermo Fisher Scientific and were based on the following sequences: hFis1 siRNA (sense 5′-GCAAGUACAAUGAUGACAUCCGUAA-3′), Mfn1 siRNA (sense 5′-GCUGGAUAGCUGGAUUGAUAAGUUU-3′), Mfn2 siRNA (sense 5′-GGACAAAGUUCUGCCCUCUGGGAUU-3′), and Dyn2 siRNA (sense 5′-AGUCCUACAUCAACACGAAtt-3′).

## Fluorescence and confocal microscopy imaging

Fluorescence and confocal microscopy imaging were performed as previously described (Zhao *et al*, 2011; Liu *et al*, 2013; Yu *et al*, 2017). Briefly, cells were plated on glass coverslips for 18 to 20 h and transfected with 0.5 μg of plasmid DNA. For mitochondrial staining, 500 nM MitoTracker Red CMXRos was added into culture medium and incubated with cells for 15 min before fixation. Cells on glass coverslips were fixed for 10 min at room temperature with 4% paraformaldehyde and permeabilized by 0.05% Triton X-100. After blocking with 1% BSA, specimens were immunostained with the indicated antibodies, mounted with the mounting medium containing DAPI for staining of nuclei (Vector Laboratories), and examined under the TCS SP5 confocal microscopy system (Leica). Quantitative co-localization analysis of confocal images for measuring the extent of mitochondrial fusion was carried out with the Pearson's correlation coefficient (r) of the Leica integrated program. Quantitative analyses of the size and number per cell of mitochondria used ImageJ software (Particle analysis).

## Co-immunoprecipitation (co-IP) and Western blot

Co-IP experiments were performed as previously described (Hajek *et al*, 2007; Zhao *et al*, 2011; Yu *et al*, 2017). For chemical crosslinking followed by co-IP, cultured cells were washed in PBS buffer and subjected to *in vivo* chemical crosslinking with 1% formaldehyde (FA) in PBS buffer, and the crosslinking reaction was stopped by washing in PBS containing 100 mM glycine. Cells were suspended and sonicated in lysis buffer (PBS containing 1% NP-40 and protease inhibitor cocktail complete EDTA-free, Roche Diagnostics). Cell lysates were used for co-IP after centrifugation. For the co-IP of endogenous proteins, Dynabeads protein G (Invitrogen) were incubated with 2 μg of antibody against target proteins at room temperature for 1 h and washed twice with lysis buffer, and then incubated with cell lysates overnight at 4°C. The immunocomplexes bound to the beads were dissolved in SDS sample buffer and detected by Western blot. For co-IP without crosslinking, cells were suspended in lysis buffer directly, and then, co-IP was performed as described above.

For Western blot, proteins were separated on NUPAGE™ 4–12% Bis–Tris protein gels (Invitrogen) and transferred to PVDF membranes using Trans-Blot Turbo transfer system (Bio-Rad). Membranes were blocked with 10% nonfat milk in PBS and incubated with the indicated primary antibodies followed by the mouse or rabbit peroxidase-conjugated secondary antibody (GE Healthcare). Blots were detected with the Pierce ECL Western Blotting Substrate (Thermo Scientific).

## PEG mitochondrial fusion assay

Polyethylene glycol (PEG)-mediated cell fusion assay was performed as described (Liesa *et al*, 2008; Kamp *et al*, 2010; Zhao *et al*, 2011). Briefly, for Myc-hFis1 plasmid transfection, cells with stable expression of mitoRFP or mitoGFP were mixed (1:1) and co-cultured for 16 h on glass coverslips, and then transfected with Myc-hFis1 plasmid or with empty vector as control for 18 h. Cycloheximide (20 μg/ml, Sigma) was added in DMEM without serum 30 min before cell fusion to inhibit the proteins synthesis. PEG-mediated cell fusion was performed by treatment with 500 μl of a pre-warmed (37°C) 50% (wt/vol) PEG 1500 (Sigma) solution in DMEM without serum for 90 s and washed three times with DMEM containing 10% serum for 10 min per wash, followed by incubation in medium with cycloheximide (20 μg/ml) at 37°C for 5 h to maintain inhibition of protein synthesis. Cells on glass coverslips were fixed, permeabilized, and blocked as described before for immunofluorescence microscopy. After immunofluorescence staining with anti-Myc tag primary antibody and DyLight 649-conjugated secondary antibody for Myc-hFis1, polykaryons (containing only two nuclei) were analyzed under the SP5 confocal microscopy system (Leica).

For hFis1 siRNA treatment followed by cell fusion assay, the cells with mitoRFP or mitoGFP were first treated with siRNA twice as described above. After 48 h of siRNA treatment, cells with mitoRFP and mitoGFP were mixed (1:1) and co-cultured on glass coverslips for another 24 h; then, the mixed cells on coverslips were subjected to PEG-mediated cell fusion as described above. At the same time, some of the cells treated with siRNA were kept in culture for 72 h after siRNA treatment and harvested for testing the extent of hFis1 knockdown by Western blot.

## Mito-PAGFP-based mitochondrial fusion assay

The mito-PAGFP-based assay was performed as previously described (Karbowski *et al*, 2004, 2014). For hFis1 overexpression, WT and Drp1$^{-/-}$ 293T cells were co-transfected with mito-PAGFP (0.5 μg), mito-DsRed (0.2 μg), and either empty vector (0.5 μg) or

Myc-hFis1 plasmid (0.5 μg). For hFis1 knockdown, WT and Drp1$^{-/-}$ 293T cells were treated with scrambled (control) siRNA or hFis1 siRNA for 48 h as described above and then co-transfected with mito-PAGFP (0.5 μg) and mito-DsRed (0.2 μg). The next day (~16 h posttransfection), images were captured on the Leica TCS SP5 confocal microscope using a 63× oil objective. Transfected cells were identified as containing red fluorescent mitochondria. A circular region of interest (ROI) with a 3 μm diameter was selected and photoactivated by a single-pulse 405 nm laser. Sequential *Z*-stack images with red and green fluorescent signals were acquired before and immediately following activation at 40 s, 15, 30, and 45 min using series of Z-sections from the top to the cell bottom with intervals between sections set to 0.5–0.75 μm. Images show *Z*-stack reconstruction of representative cells at each time point. Five to ten cells were assayed per condition. Quantitative analysis of changes in fluorescence intensity of photoactivated mito-PAGFP in ROIs at each time point was carried out using the software of the Leica Confocal System and normalized by the mito-PAGFP fluorescence at 40 s after photoactivation (as 100%).

### GTPase activity assay

The GTP hydrolysis activity assay was carried out as previously described (Stafa *et al*, 2012; Wang *et al*, 2012; Biosa *et al*, 2013) by measuring the release of free r-phosphate (Pi) from GTP using the colorimetric ATPase/GTPase activity assay kit (Sigma-Aldrich). Briefly, 293T cells were transiently transfected with Mfn1-Myc, Mfn2-Myc, Mfn2$^{K109A}$-Myc, or OPA1-Myc for 20–22 h, washed twice with phosphate-free buffer (10 mM Tris–HCl pH 7.5, 150 mM NaCl), and lysed in phosphate-free lysis buffer (10 mM Tris–HCl pH 7.5, 150 mM NaCl, 1% NP-40, protease inhibitor cocktail complete EDTA-free, Roche Diagnostics) on ice for 1 h. After centrifugation to remove insoluble debris, cell lysates were subjected to immunoprecipitation (IP) with 40 μl of anti-Myc tag agarose bead suspension (Abcam) and kept rotating at 4°C overnight. Also, 293T cells transfected with empty vector were used as control for IP. After stringent washing three times with lysis buffer, agarose beads were incubated with or without 25 nM of recombinant hFis1 protein (Abnova) or recombinant GST protein (Sigma) as negative control in lysis buffer at room temperature for 2 h, washed thrice with lysis buffer and twice with 0.5M Tris–HCl pH7.5, and finally re-suspended in assay buffer provided with the kit and subjected to GTPase activity assay according to the manufacturer's protocol. Anti-Myc tag agarose beads incubated with the cell lysate from 293T cells transfected with empty vector were used as the mock control of the GTPase activity assay.

For measuring the GTP hydrolysis activity of Drp1 and Dyn2, Drp1$^{-/-}$ 293T cells overexpressing untagged Drp1 or Drp1$^{Q34A}$ and endogenous Dyn2 in Drp1$^{-/-}$ 293T cells were used for IP and GTPase hydrolysis assay as described above. Briefly, cell lysates were subjected to immunoprecipitation with Dynabeads protein G beads (Invitrogen) overnight at 4°C (the beads had been pre-incubated with 4 μg of antibody against Drp1 or Dyn2 at room temperature for 1 h). Immunopurified untagged Drp1, Drp1$^{Q34A}$, and endogenous Dyn2 were used for testing the GTP hydrolysis activity of Drp1 and Dyn2 in the presence or absence of recombinant hFis1 or GST, respectively. Normal mouse and rabbit IgG were used as controls for co-IP and the GTPase activity assay. To assess the input levels of immunopurified proteins used in GTPase activity assay, the immunoprecipitates were subjected to Western blot analysis with antibodies as indicated.

### Statistical analysis

The difference between experimental groups was analyzed by the Student's *t*-test online software (http://www.physics.csbsju.edu/stats/t-test.html). A *P*-value < 0.05 was considered statistically significant. The means and standard errors of the mean (SEM) were calculated by a web-based statistics software (http://www.endmemo.com/math/sd.php).

**Expanded View** for this article is available online.

### Acknowledgements

The majority of this work was financed by grants from the Swedish Research Council VR-NT and additionally by VR-MH and VR-Linné. It was also supported by grants from the Swedish Cancer Society, the Cancer Society in Stockholm, Karolinska Institutet, and the Chinese Scholarship Council. We would like to thank the Knut and Alice Wallenberg Foundation (CLICK facility) for supporting this work and Dr. Florian Salomons for helpful advice on live imaging experiments.

### Author contributions

JZ and MN conceived, designed, and supervised the work. RY and JZ performed most of the experiments and analyzed data. RY, S-BJ, and JZ performed CRISPR/Cas9-based gene knockout and analyzed data. RY and JZ prepared figures. JZ, MN, UL, and RY wrote the manuscript. All authors reviewed the manuscript.

### Conflict of interest

UL holds research grants from Merck AG and AstraZeneca, no personal remuneration. The other authors declare no competing financial interests.

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
