## [Review Process File · The EMBO Journal]

Human Fis1 regulates mitochondrial dynamics through inhibition of the fusion machinery

Rong Yu, Shao-Bo Jin, Urban Lendahl, Monica Nistér and Jian Zhao.

Review timeline:

Submission date:	22 nd May 2018
Editorial Decision:	6 th July 2018
Editorial Correspondence	10 th July 2018
Revision received:	2 nd December 2018
Editorial Decision:	24 th January 2019
Revision received:	25 th January 2019
Accepted:	29 th January 2019

Editor: Elisabetta Argenzio

Transaction Report:

1st Editorial Decision

6th July 2018

Thank you for submitting your manuscript on a role for Fis1 as a mitochondrial fusion inhibitor to The EMBO Journal. Three referees have originally been assigned to your manuscript but we have now received only two reports on your study, which are enclosed below for your information. We are still expecting the comments from the last referee, which we hope we might be able to receive soon. In the meantime, we made our decision based on the two reports we already have.

As you can see, while both referees consider the findings interesting, they also raise critical points that need to be addressed before they can support publication at The EMBO Journal. In particular, both referees request you to show whether or not the mitochondria fragmentation phenotype observed upon Fis1 expression is an overexpression artifact. Referee #1 also finds important that you test: i) the effects of Fis1, Mfn1/2 and OPA1 co-expression; and ii) whether Fis1-induced mitochondria fragmentation depends on OPA1 and/or Mfn1/2. Also, this referee points out that reduced mitochondrial fusion upon Fis1 overexpression has to be confirmed using different assays. Referee #2 requests you to address the question as to how blockade of fusion results in mitochondria fragmentation in absence of fission activity, as well as to cite and discuss the recent conflicting literature.

Addressing these issues through decisive additional data as suggested by the referees would be essential to warrant publication in The EMBO Journal. Given the overall interest of your study, I would thus like to invite you to revise the manuscript in response to the referee reports.

REFeree REPORTS

Referee #1:

The study by Yu et al. describes an unexpected novel mechanism of the role of the mammalian fission factor hFis1 in regulating mitochondrial dynamics. In contrast to the prevailing view as a factor recruiting other fission factors, the authors suggest that hFis1 mediates mitochondrial fragmentation by hindering mitochondrial fusion in a DRP1- and Dyn2-independent manner. Therefore, a novel role of Fis1 as a fusion inhibitor, distinct from its role in baker's yeast, is proposed and supported by a number of experiments. There is no doubt that this work is of high general interest and the manuscript is well written. The experiments are mostly of high quality and largely demonstrate what is claimed. Still, given that this study is contrary to the prevailing view and thus fundamental for our understanding of how mitochondrial fusion is regulated in mammals, I think that some important experiments/controls are required prior to publication.

Major Points:

- 1) It is well demonstrated that overexpression of Fis1 leads to fragmented mitochondrial morphology consistent with earlier studies. Still, this could be an indirect effect caused simply by overexpression. The authors should rule out that the mitochondrial fragmentation they observe is an overexpression artifact. One suggestion is to express a mistargeted or inactive variant of Fis1 as a control.
- 2) The authors show that recombinant hFis1 reduces GTPase activities of Mfn1/2 and OPA1 (Figure 5) which could explain hFis1-induced mitochondrial fragmentation. This experiment is lacking a control in which another recombinant protein or inactive variant of hFis1 or at least GST alone is used. The control showing that GTPase activities of Drp1 or Dyn2 are not affected is not sufficient here. Impurities or detergents (?) from the protein purification could specifically affect GTPases of the tested fusion factors.
- 3) What happens to mitochondrial morphology when you overexpress Fis1 and Mfn1/2/OPA1 at the same time? Is the mitochondrial morphology fragmented or tubular? Is Fis1 functioning upstream of Mitofusins and OPA1?
- 4) Related to this, an important aspect that needs to be addressed is whether hFis1-induced mitochondrial fragmentation indeed depends on OPA1 and/or Mfn1/2. A suggestion is to overexpress hFis1 in DRP1/OPA1 DKO cells and assaying for morphology changes and mitochondrial fusion (in a way as done in Fig. 6E and 3E for DRP1 KO cells). This will tell whether hFis1-dependent fragmentation depends on OPA1. I see that a similar approach for Mfn1 is difficult to do as here mitochondria are already fully fragmented (which is not the case for the DRP1/OPA1 DKO cells as shown in Fig. 6E).
- 5) The authors use the PEG cellular fusion assay to show that Fis1 overexpression leads to reduced mitochondrial fusion (Fig. 4). As this one of the main messages of this paper another assay is needed in my opinion as well. A photoactivatable GFP-based assay should be used for this purpose as it has the major advantage of quantifying the relative rates of mitochondrial fusion in a time-dependent manner (e.g. the authors could use 15, 30 and 45 min to compare different conditions).

Minor points:

- 6) In Figure 3 the authors show that depletion Fis1 in DRP1 KO cells leads to formation of mitochondrial clusters. What is the cluster made up of? Does this represent fragmented mitochondria which cannot be spatially resolved or tubular mitochondria which are connected/fused. Photoactivation assay will help to resolve this question better as well.
- 7) Negative controls for the IPs must always be shown (Fig. 3E-K) such as the GAPDH control not only in the input but also in the IP elution.
- 8) The pull-down in Fig. 3F is not convincing as in the control IgG lane there are also weak bands for OPA1 and Mfn1 and Mfn2 (see also point 6.).
- 9) The last main figure may appear in the supplement as the role of actin in this story of mitochondrial fragmentation is out of context but still is interesting.
- 10) The authors performed the GTPase activity in figure 5 in Drp1 KO cells instead of WT cells. Why so? A valid reason should be given.

Referee #2:

The yeast gene FIS1 (fission 1) was discovered in a screen for proteins involved in mitochondrial fission. Fis1 is a tail anchored protein of the outer membrane that binds to soluble adaptor proteins to recruit a dynamin-related protein to mitochondrial fission sites. The role of its human homolog, hFis1, is less clear. Homologs of the adaptor proteins are lacking in higher eukaryotes, and there are several alternative outer membrane receptors for the recruitment of dynamin-related proteins to fission sites. Conflicting results exist in the literature about the role of hFis1 in mitochondrial division. Thus, to understand the molecular machinery of mitochondrial dynamics and its regulation in mammalian cells it would be important to resolve the role of hFis1. In the present study, the authors propose that hFis1 does not directly interact with the mitochondrial fission machinery, but inhibits the mitochondrial fusion machinery. This is certainly an interesting hypothesis, and the authors report a number of interesting observations. However, I'm not entirely convinced that their data fully support their conclusions.

Major concerns

1. It is sometimes observed that overexpression of mitochondrial outer membrane proteins results in fragmentation of the mitochondrial network (unfortunately, such experimental artifacts often remain unpublished). It is essential that the authors show that the observed fragmentation of mitochondria is not just an artifact of hFis1 overexpression. They should overexpress a non-functional hFis1 variant (e.g. a GFP fused to the hFis1 membrane anchor) to the same level and show that these cells have wild type-like mitochondrial morphology.

2. The authors try to completely block mitochondrial fission by Dyn2-siRNA in Drp1-ablated cells. Surprisingly, mitochondria still fragment upon hFis1 overexpression. How can this be explained? In analogy to published yeast experiments one might expect to find tubular, but non-dynamic mitochondria in cells completely lacking any fusion and fission activity. It is difficult to see how a block of fusion could result in mitochondrial fragmentation in the absence of any fission activity. This is a central point of this study and needs to be addressed.

3. The authors estimate mitochondrial fusion efficiencies in Fig. 4. However, fusion efficiencies of tubular mitochondria, fragmented mitochondria, or fused mitochondrial networks cannot be directly compared. For fused mitochondrial networks, a single fusion event would be sufficient to allow complete content mixing, whereas hundreds of fusion events would be required in cells with fragmented mitochondria (according to Fig. 1D wild type cells already have more than 200 mitochondria per cell). The observations shown in Fig. 4 can be explained simply by the fact that fusion of interconnected mitochondria leads to more efficient content mixing than fusion of fragmented mitochondria. Therefore, it is not possible to draw any conclusions on the activity of the fusion machinery.

4. A recent study by Wong et al. (Nature 554:382-386) reports conflicting results suggesting that hFis1 is involved in the regulation of mitochondria-lysosome contacts, and that this activity promotes mitochondrial fission. This study should be cited and thoroughly discussed.

Further concerns

5. The CoIP experiments (Fig. 3) are not entirely convincing. When hFis1 interactions with fission components are compared with fusion components, all samples should be from the same experiment (e.g. Figs. 3D and 3E). I don't see how hFis1 could interact with OPA1. Is there a residue in the intermembrane space that could be cross-linked to OPA1? In Fig. 3F Mfn1, Mfn2, and OPA1 signals can be clearly seen in the negative control lane. In Fig. 3G Dyn2 seems to interact with hFis1.

6. I could not find the information what the negative control in the GTPase activity assay (Fig. 5A) is. Furthermore, when fusion and fission factors are compared, all samples should be analyzed in the same experiment (Figs. 5A, C, E).

7. In contrast to Fig. 1 the experiment shown in Fig. 6 was performed in the DYN2 wild type background. It cannot be excluded that mitochondrial fragmentation is due to Dyn2 activity.

Additional points

8. The authors argue in the introduction (p. 3) that the machinery regulating mitochondrial fission and fusion should have a monophyletic origin and thus be conserved across all species because mitochondria were derived from a single endosymbiosis event. They should consider that the mitochondrial division machinery evolved much later. The endosymbiont used FtsZ-like division proteins that were replaced by dynamin-related proteins relatively late in evolution. There are still some unicellular algae and slime molds that have FtsZ-related mitochondrial division proteins.

9. The authors frequently use the term 'cytoplasm' instead of the correct term 'cytosol' ('cytoplasm' is the part of the cell outside the nucleus, i.e. mitochondria are part of the cytoplasm).

Editorial Correspondence

10th July 2018

As already outlined last week in the decision letter, I have now received referee #3 report on your manuscript, which is enclosed below for your information.

As you can see, referee #3 largely agrees with the other two reviewers. While this referee finds the topic of your study interesting, however s/he is concerned about the physiological relevance of the findings (point 1, 5 and 7) and requests you to employ alternative assays to prove that Fis1 overexpression reduces mitochondrial fusion. In addition, this referee stresses the lack of important controls in support of the main findings (point 2 and 6).

I would then invite you to revise the manuscript in response to all the referee reports and ask you to address these issues together with those pointed out by the other two referees.

 REFEREE REPORT.

Referee #3

The authors propose a very interesting hypothesis where Fis1 is blocking fusion. The images are beautiful and the writing is clear.

Main concerns:

Throughout the manuscript there is a general misconception that fragmentation equals a reduction in fusion. Fragmentation may be the combination of both effects on fusion and fission. There are many indirect ways in which fusion can be completely arrested, leading to fragmentation. The toxic overexpression of a protein and its aggregation is one such mechanism.

Previous studies have shown that DRP1-Fis1 interaction is not essential for fission, but that this interaction is playing a role in mitophagy related fission and in autophagosome formation.

Specifically,

1. In Fig. 1 the authors show that the over expression of Fis1 lead to fragmentation. This is based entirely on over expression which is entirely artificial. The question is whether this is physiological. What is the level of Fis1 before and after the over expression?

Later in the manuscript the authors show that lantruculin b partially inhibit this process. This will work if Fis1 connects with the ER. How this works?

2. In Fig 3 the authors show a co-IP. However there should be another negative control with another outer membrane protein. The reason being that given the large over expression levels it is likely that Fis1 is coating the outer membrane. Therefore, one should control for the possible non-specific binding to other outer membrane proteins.

3. In Fig. 3 the claim that Drp1 does not bind Fis1 is incorrect. It does when mitophagy is induced.

4. The PEG assays are over interpreted that there is a change in the rate of fusion events. The same PEG result can come from a change in fission events. An assay that measure the number of fusion events per time is needed.

5. Fig 4 the effect of Fis1siRNA could be due to a change in fission. The over expression of Fis1 gives less fusion and fragmentation, but this seems non physiological since the cells look to be in a

state of no fusion at all as expected from toxicity. A dose response of Fis1 over expression will be helpful here.

6. In Fig. 5 the authors look at Fis1 capacity to block GTPase activity. There are other proteins with GTPase that can contaminate this assay. One needs a negative control consisting of a Fis1 mutant that is lacking the GTPase domain. A second missing control is an Mfn2 that is lacking the GTPase activity.

7. The solubility of full length Fis1 is questionable. Fis1 full length is expected to denature and form aggregate due to the hydrophobic tail.

8. OPA1 is not on the outer membrane. How is it inhibited by Fis1 ?

1st Revision - authors' response

2nd December 2018

Point-by-point response to the reviewers' comments:

Referee #1:

The study by Yu et al. describes an unexpected novel mechanism of the role of the mammalian fission factor hFis1 in regulating mitochondrial dynamics. In contrast to the prevailing view as a factor recruiting other fission factors, the authors suggest that hFis1 mediates mitochondrial fragmentation by hindering mitochondrial fusion in a DRP1- and Dyn2-independent manner. Therefore, a novel role of Fis1 as a fusion inhibitor, distinct from its role in baker's yeast, is proposed and supported by a number of experiments. There is no doubt that this work is of high general interest and the manuscript is well written. The experiments are mostly of high quality and largely demonstrate what is claimed. Still, given that this study is contrary to the prevailing view and thus fundamental for our understanding of how mitochondrial fusion is regulated in mammals, I think that some important experiments/controls are required prior to publication.

We thank the reviewer for the positive comments on the novelty and quality of the presented data.

Major Points:

1) It is well demonstrated that overexpression of Fis1 leads to fragmented mitochondrial morphology consistent with earlier studies. Still, this could be an indirect effect caused simply by overexpression. The authors should rule out that the mitochondrial fragmentation they observe is an overexpression artifact. One suggestion is to express a mistargeted or inactive variant of Fis1 as a control.

This is a very good suggestion and we have approached this in two ways: by using mutated versions of Fis1 and by addressing whether apoptosis or autophagy was observed. First, we used several mutated versions of hFis1, including hFis1^{DTM/C} (lacking the TM domain and the C-terminal tail; i.e. a cytosolic hFis1), GFP-hFis1^{D1-121} (lacking the whole cytosolic domain of hFis1; i.e. a hFis1 membrane anchor linked to GFP), hFis1^{K149/K151A} (an ER mistargeted mutant; i.e. a mutant carrying lysine to alanine mutations in residues 149 and 151) and finally hFis1/yTom5C (the TM and C-tail domains of hFis1 were replaced with the corresponding regions of the mitochondrial C-tail-anchored yeast protein Tom5) to address this possibility (see also comments from referee #2 below). The results demonstrate that these hFis1 mutants significantly reduced or abolished mitochondrial fragmentation in both WT and Drp1 KO 293T cells, supporting the view that hFis1-induced mitochondrial fragmentation is not an overexpression artifact. These data are presented in the new Fig. 1D and E (see also new Fig. EV2A) in the revised version.

Second, we assessed the possibility that hFis1 overexpression may induce apoptosis and autophagy. We show that overexpression of hFis1 and mutants did not induce apoptosis and autophagy as analyzed by Western blotting of the cleavage of PARP and LC3B (new Fig. EV2B and C), suggesting that the hFis1-induced fragmentation is not a result of apoptosis or autophagy, but is indeed a specific effect.

2) The authors show that recombinant hFis1 reduces GTPase activities of Mfn1/2 and OPA1 (Figure 5) which could explain hFis1-induced mitochondrial fragmentation. This experiment is lacking a control in which another recombinant protein or inactive variant of hFis1 or at least GST alone is used.

The control showing that GTPase activities of Drp1 or Dyn2 are not affected is not sufficient here. Impurities or detergents (?) from the protein purification could specifically affect GTPases of the tested fusion factors.

This is a valid point which was also raised by the two other reviewers, and we recognize that the original set of data did not fully address this possibility. In the revised version, we have used GST protein as control to address this. The data show that that GST protein did not affect the GTPase activity, whereas GST-hFis1 clearly impaired the GTPase activity of Mfn1, Mfn2 and OPA1. The new data are presented in new Fig. 7A of the revised manuscript.

To exclude potential effects of impurities and detergents during the purification of Drp1 and Dyn2, we have in parallel used a GTPase-deficient Drp1 mutant (Drp1^{Q34A}) as control during immunoprecipitation (see new Fig. 7C).

3) What happens to mitochondrial morphology when you overexpress Fis1 and Mfn1/2/OPA1 at the same time? Is the mitochondrial morphology fragmented or tubular? Is Fis1 functioning upstream of Mitofusins and OPA1?

We thank the reviewer for raising this important question. Accordingly, we have tested potential effects of Mfn1/2 and OPA1 on the hFis1 overexpression-induced mitochondrial fragmentation. The new data show that the combined expression of hFis1 and one of pro-fusion GTPases (Mfn1, Mfn2 and OPA1), especially Mfn2, partially decreased the hFis1 overexpression-induced mitochondrial fragmentation in Drp1 KO cells. This allows us to further demonstrate that Mfn1, Mfn2 and OPA1 are important in regulating mitochondrial fission phenotype induced by hFis1 overexpression. The profusion proteins seem to inhibit/compete with the hFis1-induced fragmentation effect. This is consistent with the proposed view of hFis1 as an inhibitor of mitochondrial fusion. The data have been added as new Fig 7G and Fig. EV4 in the revised version.

4) Related to this, an important aspect that needs to be addressed is whether hFis1-induced mitochondrial fragmentation indeed depends on OPA1 and/or Mfn1/2. A suggestion is to overexpress hFis1 in DRP1/OPA1 DKO cells and assaying for morphology changes and mitochondrial fusion (in a way as done in Fig. 6E and 3E for DRP1 KO cells). This will tell whether hFis1-dependent fragmentation depends on OPA1. I see that a similar approach for Mfn1 is difficult to do as here mitochondria are already fully fragmented (which is not the case for the DRP1/OPA1 DKO cells as shown in Fig. 6E).

This is a good point and we thank the reviewer for suggesting this experiment. In the revised version, we exogenously express hFis1 in Drp1/OPA1-deficient cells. The data show that overexpression of hFis1 in Drp1/OPA1 DKO cells results in a fragmented mitochondrial phenotype, similar to that observed upon knockdown of Mfn1 or both Mfn1 and Mfn2 in Drp1/OPA1 DKO cells. This result is in line with the notion that hFis1 induces mitochondrial fragmentation by inhibition of OPA1 and Mfn1/2. The data are presented in the new Fig. 8C and D in the revised version.

5) The authors use the PEG cellular fusion assay to show that Fis1 overexpression leads to reduced mitochondrial fusion (Fig. 4). As this one of the main messages of this paper another assay is needed in my opinion as well. A photoactivatable GFP-based assay should be used for this purpose as it has the major advantage of quantifying the relative rates of mitochondrial fusion in a time-dependent manner (e.g. the authors could use 15, 30 and 45 min to compare different conditions).

We appreciate the suggestion to use an alternative assay in addition to the PEG-mediated fusion assay. To this end, we have as suggested used the photoactivatable GFP-based fusion assay as previously described (Karbowski et al., 2004; Karbowski et al., 2014). Consistent with the data from the PEG-based fusion assay, the mito-PAGFP-based fusion assay shows that overexpression of hFis1 reduces the rate of mitochondrial fusion, whereas knockdown of hFis1 by siRNA conversely increases the rate of mitochondrial fusion in both WT and Drp1 KO 293T cells. The data are presented in the new Fig. 5C, D, G and H for WT 293T and the new Fig. 6C, D, G and H for Drp1 KO cells in the revised version.

Minor points:

6) In Figure 3 the authors show that depletion Fis1 in DRP1 KO cells leads to formation of mitochondrial clusters. What is the cluster made up of? Does this represent fragmented mitochondria which cannot be spatially resolved or tubular mitochondria which are connected/fused. Photoactivation assay will help to resolve this question better as well.

This point is well taken, and we have re-examined this experiment by confocal microscopy. In fact, mitochondria appear as a tubular network in Drp1 KO cells, and there are no data showing that depletion of Fis1 can induce mitochondrial fragmentation under these conditions. In order to clearly illustrate the tubular cluster phenotype, i.e. clusters made up of tubular mitochondria, in Drp1 KO cells depleted of hFis1, we have exchanged the confocal image of hFis1 depletion in Drp1 KO cells in Fig. 3A (lower image). In the revised Figure with better resolution, it is clear that the mitochondrial clusters induced by depletion of hFis1 in Drp1 KO cells are made up of tightly packed tubular mitochondria. We also refer the reviewer to the PEG fusion assay in Fig. 6E and the photoactivation assay in Fig. 6G in the revised manuscript, in which the clustered mitochondria are clearly tubular.

7) Negative controls for the IPs must always be shown (Fig. 3E-K) such as the GAPDH control not only in the input but also in the IP elution.

We agree that data from key negative control experiments were not presented in the original manuscript and apologize for this omission. In the revised version, we have redone all co-IPs in original Fig. 3 at endogenous levels instead of using exogenous hFis1, and the GAPDH control is now included in all co-IPs (new Fig. 3D-J).

8) The pull-down in Fig. 3F is not convincing as in the control IgG lane there are also weak bands for OPA1 and Mfn1 and Mfn2 (see also point 6).

We have performed new pull-down experiments to address this, and the background in the control IgG lane is no longer visible after washing with lysis buffer thoroughly (see new Fig. 3E).

9) The last main figure may appear in the supplement as the role of actin in this story of mitochondrial fragmentation is out of context but still is interesting.

We appreciate this suggestion, and have moved the actin Figure to the supplement as new Fig. EV5.

10) The authors performed the GTPase activity in figure 5 in Drp1 KO cells instead of WT cells. Why so? A valid reason should be given.

We apologize for not making the rationale for this experiment clear. We conducted the experiment in Drp1 KO cells because we used an untagged Drp1 plasmid (Drp1 isoform 1, 736aa) expressed in cells and needed to exclude potential interference from endogenous Drp1. In the original version we performed all GTPase activity in Drp1 KO cells. We have however now repeated the GTPase activity assays for Mfn1, Mfn2, OPA1 in WT 293T cells instead of in Drp1 KO cells (new Fig. 7A) in the revised version. The GTPase activity assays for Drp1 and Drp1^{Q34A} mutant are as originally presented (new Fig. 7C) and for Dyn2 we used endogenous Dyn2 in Drp1 KO cells (new Fig. 7E). The manuscript text has been modified accordingly (page 12-13).

Referee #2:

The yeast gene FIS1 (fission 1) was discovered in a screen for proteins involved in mitochondrial fission. Fis1 is a tail anchored protein of the outer membrane that binds to soluble adaptor proteins to recruit a dynamin-related protein to mitochondrial fission sites. The role of its human homolog, hFis1, is less clear. Homologs of the adaptor proteins are lacking in higher eukaryotes, and there are several alternative outer membrane receptors for the recruitment of dynamin-related proteins to fission sites. Conflicting results exist in the literature about the role of hFis1 in mitochondrial division. Thus, to understand the molecular machinery of mitochondrial dynamics and its regulation in mammalian cells it would be important to resolve the role of hFis1. In the present study, the authors propose that hFis1 does not directly interact with the mitochondrial fission machinery, but inhibits the mitochondrial fusion machinery. This is certainly an interesting hypothesis, and the authors report a number of interesting observations. However, I'm not entirely convinced that their data fully support their conclusions.

We appreciate the positive comments on an interesting hypothesis and observations.

Major concerns

1. It is sometimes observed that overexpression of mitochondrial outer membrane proteins results in fragmentation of the mitochondrial network (unfortunately, such experimental artifacts often remain unpublished). It is essential that the authors show that the observed fragmentation of mitochondria is not just an artifact of hFis1 overexpression. They should overexpress a non-functional hFis1 variant

(e.g. a GFP fused to the hFis1 membrane anchor) to the same level and show that these cells have wild type-like mitochondrial morphology.

This is a good point, and we appreciate the reviewer's suggestion for an experiment to address this (also raised by reviewer #1, Point 1). We have expressed several hFis1 mutants including a hFis1 membrane anchor fused to GFP (see also Point 1 raised by reviewer #1). It is clear that overexpression of these mutated hFis1 proteins only marginally induced mitochondrial fragmentation, as compared to WT hFis1. The data are shown in new Fig. 1D and E (see also new Fig. EV2A) in the revised version.

2. The authors try to completely block mitochondrial fission by Dyn2-siRNA in Drp1-ablated cells. Surprisingly, mitochondria still fragment upon hFis1 overexpression. How can this be explained? In analogy to published yeast experiments one might expect to find tubular, but non-dynamic mitochondria in cells completely lacking any fusion and fission activity. It is difficult to see how a block of fusion could result in mitochondrial fragmentation in the absence of any fission activity. This is a central point of this study and needs to be addressed.

The reviewer raises an important point, and we were also puzzled by this finding. However, increasing evidence from a number of research groups show that in addition to the canonical mitochondrial fission machinery (i.e. Drp1/Dyn2-mediated fission), mitochondrial division can still occur, possibly through the cytoskeleton, the ER-mitochondria contacts and/or Fis1-dependent mitochondria-lysosome contacts in a Drp1-independent manner in mammals. In line with this reasoning, we provide evidence that F-actin filaments are involved in such a Fis1-mediated and Drp1/Dyn2-independent fission pathway (see new Fig. EV5 and last two paragraphs in the Results section). During manuscript preparation, it was reported (Wong et al. 2018, *Nature*, 554(7692):382-386) that mitochondria-lysosome contacts regulate mitochondrial fission by recruitment of Rab7-mediated GTPase-activating protein TBC1D15 to mitochondria via Fis1 to drive Rab7 GTP hydrolysis (Wong et al., 2018) in a manner independent of Drp1 (Onoue et al., 2013), and their work is also cited in the revised version. Taken together, these findings suggest that depletion of Drp1 and Dyn2 does not completely block mitochondrial fission in mammals. On the other words, mitochondrial fission can proceed via additional mechanisms in the absence of Drp1 and Dyn2. We thank the reviewer for drawing our attention to this, and we have added a discussion in the text of the revised version (pages 18-19) (see also Point 4 below).

3. The authors estimate mitochondrial fusion efficiencies in Fig. 4. However, fusion efficiencies of tubular mitochondria, fragmented mitochondria, or fused mitochondrial networks cannot be directly compared. For fused mitochondrial networks, a single fusion event would be sufficient to allow complete content mixing, whereas hundreds of fusion events would be required in cells with fragmented mitochondria (according to Fig. 1D wild type cells already have more than 200 mitochondria per cell). The observations shown in Fig. 4 can be explained simply by the fact that fusion of interconnected mitochondria leads to more efficient content mixing than fusion of fragmented mitochondria. Therefore, it is not possible to draw any conclusions on the activity of the fusion machinery.

This is a valid point, and in response to this, we have toned down the interpretation of the data on fusion efficiency (pages 10-12 in the revised version). Additionally, to further assess a possible role of hFis1 in regulating mitochondrial fusion, we have complemented the PEG-based fusion assay with a photoactivable GFP (mito-PAGFP) assay (see also referee #1, Point 5). As shown in new Fig. 5C, D and G, H as well as new Fig. 6C, D and G, H, overexpression of hFis1 reduces the rate of mitochondrial fusion, whereas depletion of hFis1 conversely increases the rate of mitochondrial fusion in both WT and Drp1 KO 293T cells as observed in the mito-PAGFP-based fusion assay. This corroborates the data from the PEG-based fusion assay.

4. A recent study by Wong et al. (*Nature* 554:382-386) reports conflicting results suggesting that hFis1 is involved in the regulation of mitochondria-lysosome contacts, and that this activity promotes mitochondrial fission. This study should be cited and thoroughly discussed.

We appreciate this comment and thank the reviewer for alerting us to this important study, which appeared during preparation of the original manuscript. We now cite the Wong et al., paper and discuss this and other studies in relation to our data (pages 18-19) in the revised version (see also Point 2 above).

Further concerns

5. The CoIP experiments (Fig. 3) are not entirely convincing. When hFis1 interactions with fission components are compared with fusion components, all samples should be from the same experiment (e.g. Figs. 3D and 3E). I don't see how hFis1 could interact with OPA1. Is there a residue in the intermembrane space that could be cross-linked to OPA1? In Fig. 3F Mfn1, Mfn2, and OPA1 signals can be clearly seen in the negative control lane. In Fig. 3G Dyn2 seems to interact with hFis1.

These are valid points and were also raised by referee #1 (Points 7 and 8). We have redone the experiments presented in previous Fig. 3D and E, so the data presented are all derived from the same experiments. The data are presented as the new Fig. 3D in the revised version.

As regards the Mfn1, Mfn2, OPA1 and Dyn2 signals in previous Fig. 3F and G, we have redone these co-IP experiments. As shown in new Fig. 3E and F, the background in the negative lane is no longer visible after more extensive washing with lysis buffer (see new Fig. 3E). The data show that hFis1 indeed specifically interacts with Mfn1, Mfn2 and OPA1, as well as with Drp1 but not with Dyn2 at endogenous levels in 293T and HeLa cells following chemical crosslinking (new Fig. 3D). Moreover, in these co-IP experiments we have added three negative control proteins including the cytosolic protein GAPDH, the MOM protein Miro1 and the MIM protein MTGM (also known as Romo1).

Finally, as shown in Fig. 3D-F, the interaction between hFis1 and OPA1 is robust in both 293T and HeLa cells at endogenous levels (see also new Fig. 3H and I), and we are also puzzled about the underlying mechanism. To explore the regions of hFis1 responsible for interaction with Mfns and OPA1, we have evaluated this using several truncated hFis1 constructs (see new Fig. 4). This clarifies that the C-terminal portion of hFis1 including the TPR2, TM domain and C-terminal tail is required for interaction with Mfns and OPA1. Moreover, the mitochondrial localization is also important because the cytosolic hFis1 mutant severely impaired these interactions. However, it is still not clear whether the C-terminal tail of hFis1 is involved in the interaction with OPA1 since the tail is also crucial for the mitochondrial localization of hFis1.

6. I could not find the information what the negative control in the GTPase activity assay (Fig. 5A) is. Furthermore, when fusion and fission factors are compared, all samples should be analyzed in the same experiment (Figs. 5A, C, E).

An appropriate control was admittedly not included in the original version and we apologize for this omission (see also response to reviewer #1, Point 2). We have now used a GST protein as negative control and also added the mock controls in these experiments (see the Methods section). Furthermore, we have used Mfn2^{K109A} and Drp1^{Q34A} as additional controls. The GTPase activity assays in the original Fig. 5 have also been redone and are now compiled in the new Fig. 7. The data show that hFis1 impairs the GTPase activity of Mfns and OPA1 but not of Drp1 and Dyn2. Since the immunopurification followed by GTPase activity assay is a time- and labor-consuming process, it is not possible to perform all these GTPase activity assays in the same experiment in our hand.

7. In contrast to Fig. 1 the experiment shown in Fig. 6 was performed in the DYN2 wild type background. It cannot be excluded that mitochondrial fragmentation is due to Dyn2 activity.

This is a good point, and we have now conducted the experiments also in a DYN2-deficient background. We found that knockdown of Dyn2 by siRNA increases the number of cells with tubular mitochondria in Drp1/OPA1 DKO cells. We also compared the mitochondrial phenotypes in Drp1/OPA1 DKO cells depleted of Mfn1 to those depleted of Mfn1 and Dyn2. Interestingly, these data show that knockdown of Dyn2 significantly decreases mitochondrial fragmentation induced by knockdown of Mfn1 in Drp1/OPA1 DKO cells. This indicates that Dyn2 is indeed involved in regulating mitochondrial division in a Drp1-independent manner when the canonical fission and fusion machineries are blocked (new Fig. 8C, lower panel and D), although the underlying mechanism remains elusive. However, the combination of Myc-hFis1 overexpression and Dyn2 siRNA does not prevent hFis1-induced mitochondrial fission (see new Fig. 8C and D).

Additional points

8. The authors argue in the introduction (p. 3) that the machinery regulating mitochondrial fission and fusion should have a monophyletic origin and thus be conserved across all species because mitochondria were derived from a single endosymbiosis event. They should consider that the mitochondrial division machinery evolved much later. The endosymbiont used FtsZ-like division

proteins that were replaced by dynamin-related proteins relatively late in evolution. There are still some unicellular algae and slime molds that have FtsZ-related mitochondrial division proteins.

We are thankful for this comment, and agree that we stretched this line of reasoning too far in the original version. In the revised version, we have amended the text in accordance with the comments (page 3).

9. The authors frequently use the term 'cytoplasm' instead of the correct term 'cytosol' ('cytoplasm' is the part of the cell outside the nucleus, i.e. mitochondria are part of the cytoplasm).

Apologies for not being stringent in terms of cytoplasm versus cytosol; we now use "cytosol" consistently.

Referee #3

The authors propose a very interesting hypothesis where Fis1 is blocking fusion. The images are beautiful and the writing is clear.

We appreciate these positive comments on our study.

Main concerns:

Throughout the manuscript there is a general misconception that fragmentation equals a reduction in fusion. Fragmentation may be the combination of both effects on fusion and fission. There are many indirect ways in which fusion can be completely arrested, leading to fragmentation. The toxic overexpression of a protein and its aggregation is one such mechanism.

Previous studies have shown that DRP1-Fis1 interaction is not essential for fission, but that this interaction is playing a role in mitophagy related fission and in autophagosome formation.

This is a valid comment. In the revised version, we have tried to be more nuanced in this regard, and not automatically equal fragmentation with reduction in fusion. We have modified the text and discussed the role of Drp1-Fis1 interaction in Drp1-mediated fission, especially in response to cell stress-triggered mitochondrial fission in the revised version (page 18). We have also addressed the potential risk for side effects of overexpressing proteins, and provide a set of appropriate control experiments to assess this question (see also responses to reviewers #1 and #2).

Specifically,

1. In Fig. 1 the authors show that the over expression of Fis1 lead to fragmentation. This is based entirely on over expression which is entirely artificial. The question is whether this is physiological. What is the level of Fis1 before and after the over expression?

This point is well taken and was commented upon also by reviewer #1 (Point 1). We have now used several mutated versions of hFis1, including hFis1^{DTM/C}, GFP-hFis1^{D1-121}, hFis1^{K149/K151A} and hFis1/yTom5C (see response to reviewer #1, Point 1 and reviewer #2, Point 1) to address this possibility. The results demonstrate that these hFis1 mutants only marginally induced mitochondrial fragmentation in WT and Drp1 KO 293T cells. This is presented in Fig. 1D and E (see also new Fig. EV2A) in the revised version. Based on these new data, we believe that the mitochondrial fragmentation phenotype observed after overexpression of WT hFis1 can be regarded as a Fis1-specific effect.

We also show the relative levels of hFis1 before and after overexpression in both WT and Drp1 KO 293T. The results are presented in new Fig. EV2B and C. In our cell experiments for observation of mitochondrial phenotypes, transient transfections were usually performed with a lower dose of plasmid DNA (0.5 mg per well in 6-well plates) and cells were harvested 18 to 20 h after transfection. Although it was difficult to control expression levels of WT hFis1 in different cells during transient expression, most of cells with exogenous expression of hFis1 showed the fragmented mitochondrial phenotype.

Later in the manuscript the authors show that lantruculin b partially inhibit this process. This will work if Fis1 connects with the ER. How this works?

We agree with the reviewer that an interaction between hFis1 and the ER could be a plausible explanation. To address this, we have generated and analyzed a hFis1 mutant (hFis1^{K149/K151A}), which has been shown to mistarget most of hFis1 to the ER (Delille and Schrader, 2008; Stojanovski et al., 2004). Compared with WT hFis1, overexpression of Myc-hFis1^{K149/151A} significantly reduced the number of cells with fragmented mitochondria in both WT and Drp1^{-/-} 293T cells. Instead, the

panel of hFis1 mutants used indicated that localization of hFis1 in the mitochondrial membrane is essential inducing the fragmented mitochondrial phenotype. This is presented in new Fig. 1D and E (see also new Fig. EV2A) in the revised version. Although additional involvement of mitochondria-ER contacts could not be further evaluated in this study due to experimental technical problems using 293T cells, it is still quite interesting to elucidate the potential role of ER in the hFis1-mediated mitochondrial fission in the future.

2. In Fig 3 the authors show a co-IP. However, there should be another negative control with another outer membrane protein. The reason being that given the large over expression levels it is likely that Fis1 is coating the outer membrane. Therefore, one should control for the possible non-specific binding to other outer membrane proteins.

We thank the reviewer for suggesting a good control experiment (see also reviewer #1, Point 7). To exclude potential artifacts due to overexpression, we have in the revised version performed all co-IP experiments at endogenous levels and added GAPDH, Miro1 and MTGM (Romo1) as negative controls. As shown in new Fig. 3D-F in the revised version, hFis1 specifically binds to Mfn1, Mfn2 and OPA1, as well as to Drp1 following chemical crosslinking, whereas no interactions are detected between hFis1 and GAPDH, Miro1 and MTGM.

3. In Fig. 3 the claim that Drp1 does not bind Fis1 is incorrect. It does when mitophagy is induced.

We are grateful for the reviewer's comments and indeed, several previous studies show that the DRP1-Fis1 interaction increases in response to cell stress-, drug treatment-triggered and mitophagy/apoptosis related mitochondrial fission. In fact, a weak interaction between Drp1 and hFis1 could be seen in the original Fig. 3D and in the new Fig. 3D following chemical crosslinking. We agree that we omitted the importance of the Drp1-Fis1 interaction in mitochondrial dynamics in the original manuscript. In the revised version we have amended the text in the Discussion section to reflect that Drp1 and Fis1 can interact, for example when mitophagy is induced (page 18).

4. The PEG assays are over interpreted that there is a change in the rate of fusion events. The same PEG result can come from a change in fission events. An assay that measure the number of fusion events per time is needed.

This is an insightful comment, also raised by reviewer 1. In the revised version, we complement the PEG-based fusion assay with a photoactivatable GFP-based fusion assay to score the fusion events at different time points (see new Fig. 5C, D and G, H in WT 293T cells and new Fig. 6C, D and G, H in Drp1 KO 293T cells). The data from the photoactivatable GFP-based assay show that hFis1 overexpression reduces while hFis1 knockdown enhances the rate of mitochondrial fusion. This is consistent with results from the PEG-based fusion assay.

5. Fig 4 the effect of Fis1 siRNA could be due to a change in fission. The over expression of Fis1 gives less fusion and fragmentation, but this seems non physiological since the cells look to be in a state of no fusion at all as expected from toxicity. A dose response of Fis1 over expression will be helpful here.

We understand the reviewer's concern over the Fis1 levels and the possibility to be toxic to the cells. However, in the PEG-based fusion assay and other confocal experiments in this work, overexpression of Fis1 was performed at a lower dose of plasmid (0.5 mg per well in 6-well plates) for transient transfection in both WT and Drp1^{-/-} cells and the effect of Fis1 overexpression was typically examined 18 to 20 h post-transfection. We now experimentally show that overexpression of Fis1 does not trigger apoptosis or autophagy (new Fig. EV2B and C).

6. In Fig. 5 the authors look at Fis1 capacity to block GTPase activity. There are other proteins with GTPase that can contaminate this assay. One need a negative control consisting of a Fis1 mutant that is lacking the GTPase domain. A second missing control is an Mfn2 that is lacking the GTPase activity.

This is a good point and was also raised by reviewer #1 (see Point 2), and we agree that the original version fell short on this. In the revised version, we have used a GTPase mutant Mfn2^{K109A} as well as a GTPase-deficient mutant Drp1^{Q34A} as controls. We also added recombinant GST protein control as suggested by referee #1 as well as mock control in these experiments. The data show that all WT GTPases exhibit a clear GTPase activity. In contrast, Mfn2^{K109A} and Drp1^{Q34A} severely impair the GTPase activity. The hydrolysis activity of the three pro-fusion GTPases (Mfns and OPA1) was impaired by treatment with hFis1 but not by GST protein, whereas the pro-fission GTPases (Drp1 and Dyn2) were not affected by hFis1 and GST. These data are presented as new Fig. 7 in the

revised manuscript. We believe these new experiments make it clear that hFis1 mainly regulates the activity of the three pro-fusion GTPases (Mfns and OPA1) rather than the pro-fission GTPases (Drp1 and Dyn2).

7. The solubility of full length Fis1 is questionable. Fis1 full length is expected to denture and form aggregate due to the hydrophobic tail.

The reviewer raises an interesting possibility. However, the commercially available full-length hFis1 with GST-tag at N-terminal (Abnova, catalog#: H00051024-P01) is soluble in the storage buffer with 50mM Tris-HCl. During our experiments, the GST-Fis1 recombinant protein was totally soluble in the buffers used, no precipitate could be observed.

8. OPA1 is not on the outer membrane. How is it inhibited by Fis1?

The interaction between hFis1 and OPA1 is robust, but we are also puzzled by the fact that OPA1 is inhibited. We believe that this may occurs via an indirect mechanism, which we now specifically discuss in the revised version in the Discussion section (page 16).

In conclusion, we thank the three reviewers for fair and constructive comments, which have been very helpful to improve the manuscript.

References

- Delille, H.K., and M. Schrader. 2008. Targeting of hFis1 to peroxisomes is mediated by Pex19p. *The Journal of biological chemistry*. 283:31107-31115.
- Karbowski, M., D. Arnoult, H. Chen, D.C. Chan, C.L. Smith, and R.J. Youle. 2004. Quantitation of mitochondrial dynamics by photolabeling of individual organelles shows that mitochondrial fusion is blocked during the Bax activation phase of apoptosis. *J Cell Biol*. 164:493-499.
- Karbowski, M., M.M. Cleland, and B.A. Roelofs. 2014. Photoactivatable green fluorescent protein-based visualization and quantification of mitochondrial fusion and mitochondrial network complexity in living cells. *Methods in enzymology*. 547:57-73.
- Onoue, K., A. Jofuku, R. Ban-Ishihara, T. Ishihara, M. Maeda, T. Koshiba, T. Itoh, M. Fukuda, H. Otera, T. Oka, H. Takano, N. Mizushima, K. Mihara, and N. Ishihara. 2013. Fis1 acts as a mitochondrial recruitment factor for TBC1D15 that is involved in regulation of mitochondrial morphology. *Journal of cell science*. 126:176-185.
- Stojanovski, D., O.S. Koutsopoulos, K. Okamoto, and M.T. Ryan. 2004. Levels of human Fis1 at the mitochondrial outer membrane regulate mitochondrial morphology. *Journal of cell science*. 117:1201-1210.
- Wong, Y.C., D. Ysselstein, and D. Krainc. 2018. Mitochondria-lysosome contacts regulate mitochondrial fission via RAB7 GTP hydrolysis. *Nature*. 554:382-386.

Thank you for submitting a revised version of your manuscript. Please accept my apologies for the delay in getting back to you with our decision due to the recent seasonal holidays. Your revised study has now been seen by the original referees whose comments are shown below.

As you will see they find that all criticisms have been sufficiently addressed and recommend the manuscript for publication. However, before we can officially accept the manuscript there are a few editorial issues concerning text and figures that I need you to address.

REFeree REPORTS

Referee #1:

The authors have nicely and thoroughly addressed the points raised previously. In particular, the data using photoactivatable GFP is convincing. I think this is a nice and important study of high general interest.

Referee #2:

The authors have addressed my previous concerns in an adequate manner. In particular, they provide evidence with new data that mitochondrial fragmentation is not a mere consequence of hFis1 overexpression, they suggest alternative mechanisms of mitochondrial division independent of dynamins, and they improved the presentation and interpretation of fusion assays. This, together with several other modifications of text and figures significantly improved the manuscript. Publication can now be recommended.

Referee #3:

Authors have properly addressed all of my comments

Corresponding Author Name: Jian Zhao and Monica Nistér

Journal Submitted to: The EMBO Journal

Manuscript Number: EMBOJ-2018-99748